# CryoEM structure of the super-constricted two-start dynamin 1 filament

Jiwei Liu[1,5], Frances Joan D. Alvarez[2,5], Daniel K. Clare[3,5], Jeffrey K. Noel[4] & Peijun Zhang [1,2,3 ✉]

Dynamin belongs to the large GTPase superfamily, and mediates the fission of vesicles during endocytosis. Dynamin molecules are recruited to the neck of budding vesicles to assemble into a helical collar and to constrict the underlying membrane. Two helical forms were observed: the one-start helix in the constricted state and the two-start helix in the super-constricted state. Here we report the cryoEM structure of a super-constricted two-start dynamin 1 filament at 3.74 Å resolution. The two strands are joined by the conserved GTPase dimeric interface. In comparison with the one-start structure, a rotation around Hinge 1 is observed, essential for communicating the chemical power of the GTPase domain and the mechanical force of the Stalk and PH domain onto the underlying membrane. The Stalk interfaces are well conserved and serve as fulcrums for adapting to changing curvatures. Relative to one-start, small rotations per interface accumulate to bring a drastic change in the helical pitch. Elasticity theory rationalizes the diversity of dynamin helical symmetries and suggests corresponding functional significance.

[1] Division of Structural Biology, Wellcome Centre for Human Genetics, University of Oxford, Oxford OX3 7BN, UK. [2] Department of Structural Biology, University of Pittsburgh School of Medicine, Pittsburgh, PA 15260, USA. [3] Electron Bio-Imaging Centre, Diamond Light Source, Harwell Science and Innovation Campus, Didcot OX11 0DE, UK. [4] Electric Ant Lab, Amsterdam, The Netherlands. [5]These authors contributed equally: Jiwei Liu, Frances Joan D. Alvarez, Daniel K. Clare. ✉email: peijun.zhang@strubi.ox.ac.uk

**D**ynamin (isoforms 1, 2, and 3) is the founding member of a ubiquitous superfamily of large GTPases and is best known for its role in catalyzing membrane scission of cargo-loaded vesicles during endocytosis[1]. These enzymes are important drug targets as they contribute to the pathophysiology of neurological disorders (e.g., Alzheimer's disease and Charcot–Marie-Tooth (CMT) neuropathy)[2], certain cancers[3], and viral infection[4], among others. Other dynamin family members are also essential to cellular functions such as cytokinesis (e.g., ADL1 and Dyn2), organelle fission or fusion (e.g., Drp1, Atl1–3, Mfn1 and 2, Vps1, Fzo1, OPA1, and Mgm1)[5], and interferon-induced resistance to pathogens (e.g., MxA, MxB, and GBP)[6].

A single polypeptide chain of dynamin contains five structurally and functionally distinct domains: (i) the GTPase domain (G domain), which binds and hydrolyzes GTP; (ii) the bundle-signaling element (BSE), which transmits the two-way signaling between the catalytic domain and Stalk domain; (iii) the Stalk domain, which provides major interfaces for oligomerization; (iv) the pleckstrin homology (PH) domain, which targets dynamin to the membrane; and (v) the proline-rich (PRD) domain, which binds SH3-containing partner proteins at the fission site[7,8]. Two hinges are present; Hinge 1 connects the BSE and the Stalk whereas Hinge 2 connects the BSE and the G domain (Fig. 1a, b). Hinge 1 appears to be flexible, a wide range of conformations have been observed among different dynamin and dynamin-like proteins (DLP)[9].

In the absence of GTP, dynamin mainly exists as a tetramer in solution. The crystal structures show that two Stalk domains form a cross dimer through the conserved interface 2. The junction between the Stalk and the BSE and the membrane facing terminals of the Stalk provide two other conserved interfaces, interface 1 and 3 respectively, to form a tetramer[7,10] (Fig. 1b). The tetramer exhibits basal GTP hydrolysis activities (Fig. 1b). Using a Stalk truncated construct (GTPase-BSE), it has been shown that GTPase-BSE exists as a monomer in the apo and GDP bound states, and dimerization between the GTPase domains (GG dimerization) could be achieved in the presence of GDP.AlF4[4–11]. For the full-length dynamin, this GG dimerization is formed when dynamin is bound to liposomes in helical configurations. As a result, the ability to hydrolyze GTP has been stimulated over a hundred times. Various cryoEM reconstructions confirmed that a GG dimer interface forms between two dynamin monomers in helical configurations, and the three conserved Stalk interfaces are vital for dynamin to polymerize into filaments[12,13]. During the cycles of GTP hydrolysis (GTPase cycles), the GG dimer forms and dissociates repeatedly, and the energies from the hydrolysis by the dynamin helical coat drives the scission of the underlying membrane[13,14].

Thereby, it is vital to understand dynamin helical assemblies to dissect the mechanisms of membrane fission. Both the one-start and the two-start dynamin helical assemblies have been found under various in vitro conditions[15]. The one-start helices reconstructed in vitro have an outer diameter around 40 nm and inner diameter (refers to the lipid tube lumen) around 7 nm, while the two-start helices reconstructed in vitro have an outer diameter around 37 nm and inner diameter around 4 nm[1,16]. Constricting membrane tubes is critical for dynamin-mediated membrane fission[17], and the 4 nm reaches the theoretical limit required for spontaneous membrane fission[16]. It thus raises the possibility that a one-start helix constricts the membrane tubes first, and a transition from a one-start helix to a two-start helix may lead to

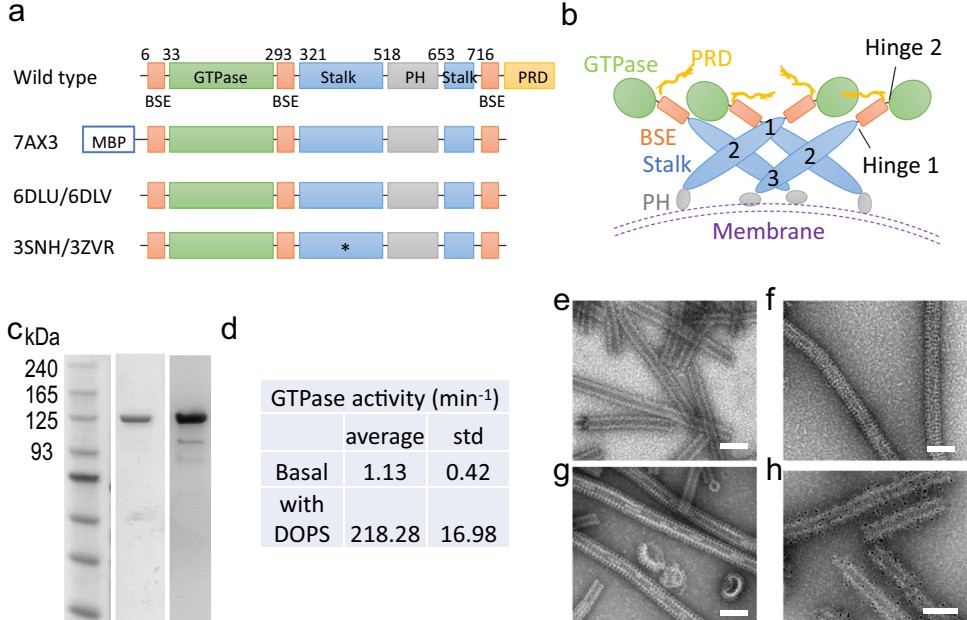

**Fig. 1 Purification and characterization of MBP-Dyn1ΔPRD. a** Schematic of dynamin 1 constructs involved in representative structural studies (PDB codes on the left). The PRD domain is unstructured, removed for most structural studies. 7AX3 is the construct used in this study, showing a two-start helix in complex with GMP-PCP in the absence of lipid. 6DLU is the one-start atomic structure, and 6DLV is the two-start low-resolution structure. * indicates assembly deficient mutant used in crystallographic studies. 3SNH: IHGIR395–399AAAAA; 3ZVR: G397D. **b** Schematic of dynamin 1 tetramer structures. The three conserved Stalk interfaces are denoted by numbers (1, 2, 3). **c** Typical SDS–PAGE (middle) and western blot (right) analysis of purified MBP-Dyn1ΔPRD. A primary antibody against dynamin 1 was used for the western blot. Three independent experiments were carried out. **d** Lipid-stimulated GTPase activity of MBP-Dyn1ΔPRD. Three independent experiments were carried out. std standard deviation. **e**–**g** Representative micrographs of negative-stained helical assemblies formed by MBP-Dyn1ΔPRD at 150 mM NaCl in the presence of GMP-PCP (**e**), DOPS (**f**), and both GMP-PCP and DOPS (**g**). **h** Immunogold labeling of MBP on MBP-Dyn1ΔPRD tubes shown in (**e**). Scale bars, 100 nm. **e**–**h** Three independent experiments were carried out.

the ultimate membrane scission. Similarly, the DLP Dnm1 mediating mitochondrial fission has also been found to exist in two-start helical configurations[18], highlighting the significance of the two-start helical forms for membrane fission. The structure of the one-start dynamin helix has been solved to 3.75 Å resolution, confirming the functional significance of the GG and the Stalk interfaces[12]. The maps of the two-start super-constricted helices are limited at ~10 Å[12], hindering the mechanistic understandings of dynamin-mediated membrane fission. Why a two-start helix exists and how a one-start constricted helix could convert into a two-start super-constricted helix remains elusive. One assumption based on the low-resolution maps of two-start helices is that the conformational changes of all or some of the interfaces (the GTPase dimeric interface, and especially the three Stalk interfaces) account for the switch of helical symmetries.

The mechanism of dynamin-mediated membrane fission has been under debate for decades[1]. Two prevailing models are emerging. In the first model, the disassembly model, dynamin forms a helical assembly that constricts the membrane and subsequent GTP hydrolysis causes disassembly leading to membrane destabilization and fission. In the second model, the constriction/ratchet model, dynamin assembles into a helical filament, where the ends are capped by dimerized GTPase domains. Subsequent rounds of GTP binding and hydrolysis lead to constriction and the eventual fission of the membrane. The main difference between these two models is how energy from GTP hydrolysis is utilized: destabilizing the membrane in the former model or constricting the membrane in the latter model.

Here, we report the structure of a two-start dynamin helix at 3.74 Å resolution. When compared to the one-start helical conformation, we show that the three Stalk interfaces are well conserved and serve as fulcrum points to allow small rotations between the Stalk domains of different monomers. A major difference resides at the Hinge 1 orientation. A relative rotation of three degrees was observed, which may provide a mechanism to transduce the energy from GTP hydrolysis to the Stalk rotations to membrane constriction. We suggest that these two conformational changes contribute to the transitioning from a one-start helix to a two-start helix. The asymmetry in Hinge 1 and Hinge 2 conformations in both the one-start and two-start helices likely supports that an active GTP hydrolysis-dependent constriction could happen in diverse helical configurations. Theoretical analysis of the filament elasticity shows that the dynamin filament favors a large pitch (two-start in this case) and stress accumulates drastically as a one-start helix constricts the underlying lipid to a narrow diameter. We discuss most existing structure solutions and propose a working model consistent with the constriction/ratchet mechanism.

## Results

**Structure determination of the two-start dynamin 1 helical assembly**. Preliminary analysis suggests that in the absence of lipids, dynamin 1 lacking the PRD domain (Dyn1ΔPRD) tends to form a two-start filament in the presence of GMP-PCP. This is probably due to the lack of lipid constraints, which allows the dynamin filament to constrict into a more compact form. However, the lipid-free dynamin tubes bundle together severely under cryogenic conditions, preventing high-resolution structure solution. We thus introduced an MBP tag fused to the N-terminus of Dyn1ΔPRD with a long linker, a strategy we used previously to determine the structure of a dynamin family member, MxB, which forms one-start helical assembly[19,20]. We expressed and purified the MBP-fused Dyn1ΔPRD protein, which is GTPase active and could tubulate lipids such as the dioleoyl phosphatidylserine (DOPS) liposomes (Fig. 1c–g). It also possesses over

100-fold stimulated GTPase activity upon DOPS binding, confirming that the MBP tag has little effect on Dyn1ΔPRD assembly and activity (Fig. 1d). The MBP tagged Dyn1ΔPRD self-assembles into long and straight two-start filaments in the presence of GMP-PCP in the absence of lipid template, same as the untagged Dyn1ΔPRD. The MBP tag is located at the outer surface of the dynamin tubes, as confirmed by negative stain EM and immunogold labeling against MBP (Fig. 1h). We collected cryoEM micrographs of the MBP tagged Dyn1ΔPRD tubes assembled with GMP-PCP (Fig. S1 and Table 1) and manually inspected the Fourier transform of the tubes to evaluate initial parameters of helical symmetry. Most tubes belong to a two-start helix with a Miller index of (−14, 2) and have outer diameters of ~37 nm (Fig. S1), consistent with the measured diameter of super-constricted dynamin lipid tubes treated with GTP[12]. After iterative rounds of 2D and 3D classifications in RELION 2.0 and 3.0, the map was refined at 3.74 Å (Fig. S2). It shows a two-start helix with a rise of 13.58 Å and a twist of 24.43°. The two rungs are related by a C2 symmetry operator (Fig. 2a and Table 1).

The map quality allows us to build an unambiguous structural model (Fig. 2c, Fig. S2, and Table 1). The three dynamin domains are radially distributed, with the G domain (Fig. 2b, d) making up the outer surface layer, the BSE in the short middle layer (Fig. 2d), connecting the G domain and the inner Stalk domain layer, which forms an extended oligomer (Fig. 2d). There is no discernible density for the MBP, which is averaged out. Neither is their density for the PH domain when reconstructions were carried out with or without applying helical symmetry. The PH domain is expected to be oriented towards the lumen of the tube, based on previous biochemical data and 3D reconstructions of lipid-bound dynamin[12], suggesting that this domain is flexible in the absence of a lipid template (Fig. 2). The density for the bound GMP-PCP nucleotide is clearly discernible (Fig. 2c, inset).

**The two Hinge conformations are asymmetric**. To allow the comparison between the one-start and two-start helix structures, we fitted the one-start GG dimer structure (PDB: 6DLU) into the corresponding one-start helix electron densities (EMD-7957) to generate a one-start helical model (Fig. 3a, top). The twist of the two-start helical assembly of dynamin is 24.43°, which corresponds to ~14.7 subunits per turn, while the twist of the one-start helix is 23.68°, corresponding to ~15.2 subunits per turn. The two-start helix comprises of two strands, which are joined by the conserved GG dimeric interfaces contributed by the GTPase domains of the two opposing strands (Figs. 2a and 3a). In the one-start helix, the GG dimer interface is formed by the GTPase domains of the dynamin cross dimers of the same strand following the completion of a helical turn (Fig. 3a).

The two monomers of the GG dimer share the same overall architecture but are not symmetrical, consistent with the GG dimer in the one-start helix[12] (Fig. 3b, c). Hinge 1 in both monomers consists of loops. When aligned on the Stalk domain, the BSE swings 12° between the two monomers, and to a larger extent 34° when compared to one monomer of the crystal structure of dynamin 3 (PDB: 5A3F) (Fig. 3d). For Hinge 2, residues I289 to P294 form a continuous helix in one monomer, but this is not the case in the other monomer, instead, there is a kink (Fig. 3b, c, e). When aligned on the BSE domain, the GTPase domain swings 46° around Hinge 2 between the two monomers (Fig. 3e). Compared to the apo crystal structure of dynamin 1 (PDB: 3SNH)[7], Hinge 2 exhibits an even larger angle (Fig. 3f). The conformations of Hinge 2 are similar between the one-start and the two-start helices (Fig. 3b, c). The triple mutant T292A/L293A/P294A, which presumably increases the helical propensity of Hinge 2, resulted in a transferrin uptake deficiency in vivo[12],

**Table 1 CryoEM data collection, refinement, and validation statistics.**

| | NT14393-32 | NT14393-34 | Combined EMD-11932 PDB-7AX3 |
|---|---|---|---|
| **Data collection and processing** | | | |
| Magnification | x130,000 | x130,000 | |
| Voltage (kV) | 300 | 300 | |
| Electron exposure (e-/Å²) | 45 | 42 | |
| Defocus range (μm) | −0.5 to −3 | −1 to −3 | |
| Pixel size (Å) | 1.048 | 1.048 | |
| Symmetry imposed | C2 helical | C2 helical | C2 helical |
| Initial particle images (no.) | 105,404 | 14,722 | 20,938 |
| Final particle images (no.) | 13,943 | 6,995 | 16,772 |
| Map resolution (Å) | 4.6 | 4.6 | 3.74 |
| FSC threshold | 0.143 | 0.143 | 0.143 |
| Map sharpening B factor (Å²) | | | −127.767 |
| Helical parameters | | | |
| rise (Å) | | | 13.58 |
| twist (°) | | | 24.43 |
| **Refinement** | | | |
| Initial model used (PDB code) | | | 6dlu |
| Model resolution (Å) | | | 3.4/3.6/3.9 |
| FSC threshold | | | 0/0.143/0.5 |
| Model composition | | | |
| Non-hydrogen atoms | | | 169717 |
| Protein residues | | | 21037 |
| Ligands | | | 36 |
| | | | GMP-PCP: 36 |
| | | | Mg: 36 |
| B factors (Å²) | | | |
| Protein | | | 76.07 |
| Ligand | | | 72.52 |
| R.m.s. deviations | | | |
| Bond lengths (Å) | | | 0.004 |
| Bond angles (°) | | | 0.753 |
| Validation | | | |
| MolProbity score | | | 2.01 |
| Clashscore | | | 12.91 |
| Poor rotamers (%) | | | 0 |
| Ramachandran plot | | | |
| Favored (%) | | | 94.35 |
| Allowed (%) | | | 5.65 |
| Disallowed (%) | | | 0 |
| Rama-Z | | | |
| whole | | | 0.87 |
| helix | | | 0.19 |
| sheet | | | 1.34 |
| loop | | | 1.61 |
| Phenix CC | | | |
| Mask | | | 0.81 |
| Volume | | | 0.80 |
| Mean for ligands | | | 0.87 |

suggesting that the kink around T292-P294 plays an important role in dynamin mediate membrane fission.

**The Hinge 1 motion.** Due to the asymmetric feature of the GG dimers in both the one-start and two-start helices, the monomers sharing similar Hinge 2 conformations are aligned when conducting the structural comparisons. The GG interface is well conserved in both one-start and two-start conformations

(Fig. S3), as well as the crystal GG dimer (with an RMSD of 0.77 between the two-start GG dimer and the crystal GG dimer). In comparison with the one-start GG dimer, there is about 3° rotation around Hinge 1 which brings the terminals of the two Stalk domains closer by 8 Å in the two-start GG dimer (Fig. 3g). This likely brings their connecting PH domains closer and rearranges the attached membrane. When the Stalk domain of one monomer is aligned between the one-start and two-start GG dimers, the BSE domain, the GTPase domain, and the Stalk domain of the other monomer would be displaced (Fig. 3h). Upon nucleotide binding and hydrolysis, the GG dimer would form and dissociate[11]. It is therefore likely that Hinge 1 communicates the conformational changes initiated at the GTPase domain to the terminal of the Stalk domain connecting the PH domain and membrane.

**The Stalk rotates around the conserved interfaces.** The helical assemblies are majorly formed via the Stalks. To understand the structural difference between the one-start and the two-start helices, we first compared the tetrameric Stalk structures from these two helices. The tetramer is composed of two layers (chain A and C in one layer and chain B and D in the other layer in Fig. 4a). When the tetramers are superimposed (chain A in Fig. 4a–e), we observed that the cross dimer interface 2 (Fig. 4a, black circles) and the oligomeric interfaces 1 and 3 (Fig. 4a, light blue and red circles, respectively) are highly conserved, despite the different helical symmetries between the two structures. The residues involved in forming all three Stalk interfaces are the same between the one- and the two-start helices (Fig. S4). Previous studies reported that point mutations of the interface residues resulted in impaired activities of dynamin in vivo and in vitro, including the abilities to hydrolyze GTP in vitro, to form filaments in vitro, and to mediate transferrin uptake in vivo (Supplementary Table 1)[7,8,10,12,21]. Because the interfaces are well conserved (Fig. S4a–c), it is worth noting that the mutagenesis analysis of the interfaces confirmed the structural solutions of both the one-start[12] and the two-start helices, as well as previous crystal structures[7,10]. Comparing the Stalk interfaces of the two-start helices with that of the crystal structure of the dynamin 3 (K361S/ΔPRD) tetramer[10], we observe that interface 2 is most conserved (Fig. S5b), while the interface 1 and 3 adopted small conformational rearrangements (Fig. S5a, S5c). The crystal structure of dynamin 3 tetramer is likely representing the solution state. Thereby, interface 2 appears most rigid, and some plasticity of interfaces 1 and 3 allows the dynamin tetramers to form a filament.

We further noticed a series of small rotations using the conserved Stalk interfaces as the fulcrums. When one of the Stalks (chain A) is aligned between the one-start and the two-start helical configurations, we observed that the other Stalk (chain D) would rotate around interface 3 (Fig. 4b). Chain C at the same layer of chain A, rotate around the interface 2 formed between chain C and chain D (Fig. 4c). This rotation is necessary to accommodate the conformational change of chain D to maintain the conserved interface 2. As a result, chain B rotates away to meet the new chain C orientation to maintain the conserved interface 1 (Fig. 4d). The movement of chain B is similar to chain D, around interface 3 (Fig. 4e). Even though chain B forms a cross dimer with chain A through the conserved interface 2, neglectable changes are observed at this interface 2 from this rotation (Fig. 4e and Fig. S4b). This likely reflects the feature of a lever-like rotation, where interface 1 is further distant than interface 2 from the fulcrum point interface 3. There appear coordinated conformational changes of each monomer in order to maintain the Stalk interfaces essential for assembly. Although the changes

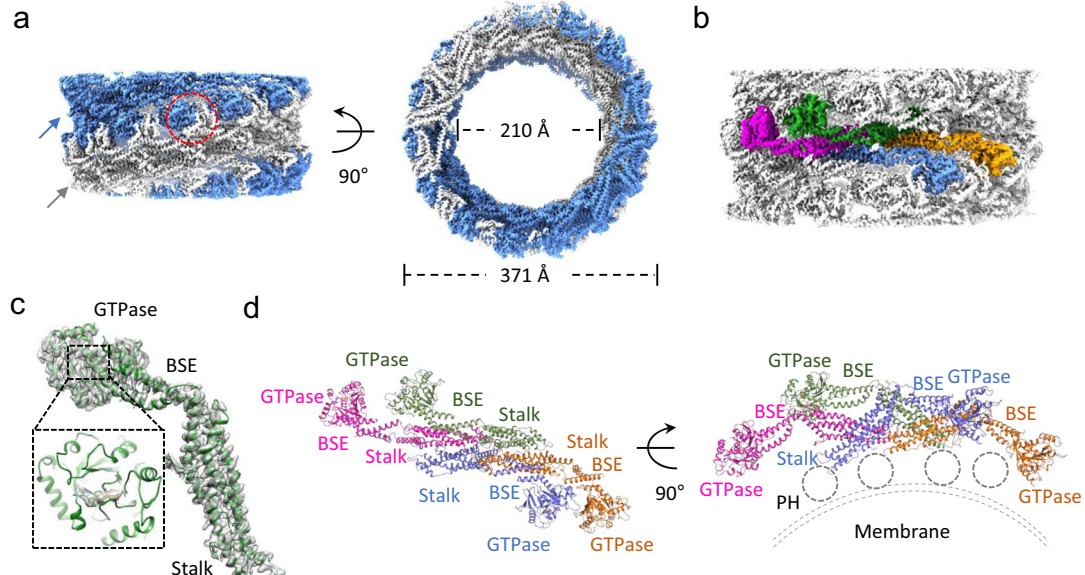

**Fig. 2 CryoEM map of the two-start dynamin 1 filament at 3.74 Å resolution. a** CryoEM density map of Dyn1ΔPRD helical assembly in the presence of GMP-PCP. The arrows indicate the two strands, blue and gray, respectively, joined by the GTP dimer interface (dashed red circle). The outer and inner diameters of the tube are shown (right). **b** A Dyn1ΔPRD tetramer is highlighted. **c** The electron densities of one monomer. Inset shows an enlarged view of the nucleotide-binding pocket (boxed region) with GMP-PCP bound. **d** The tetrameric structure of Dyn1ΔPRD, viewed from the tube surface (left) and along the tube axis (right). Each monomer is colored the same as in panel **b**. The PH domain is not visualized, shown as a dashed circle, and the membrane is drawn as gray dashed lines.

are small within a tetramer unit (Fig. 4a–e), they propagate and accumulate through multiple assembly units and lead to a drastic change of the helix (Fig. 4f).

**Two-start helix is energetically preferred**. As described above, very small alterations between successive Stalk domains are sufficient to switch between the one-start and two-start helical symmetries (Figs. 4f and 5a, b). The energetic cost of these small changes is well described by an elastic treatment. Theoretical analysis of elastic deformations in dynamin filament was explored in detail in a recent study[22] and, here, by following this analysis, we plot the elastic energy as a function of the helical parameters in Fig. 5c (see Methods for details). The elastic energy was based on the molecular dynamics simulation-derived flexibility of a single dynamin stalk tetramer, isolating the filament energy from the influence of the membrane. Interestingly, the helical parameters of experimentally characterized constricted (one-start) and super-constricted (two-start) dynamin helices lie on the minimal energy curve. Another striking result is that the dynamin filament's elastic energy is minimized at large pitches (40–50 nm).

The prediction that large helical pitches are more stable may explain the observation of multi-start helices of dynamin. Clearly, forming GG interactions between neighboring turns is an important contributor to helical shape and stability. However, we have shown that the GG dimeric interfaces between the turns are nearly identical between one- and two-start helices and, thus, can be neglected as the determining factor for filament conformation. Thus, in our simplified, membrane-free system, the Stalk filament energy becomes the dominant contributor to the helical shape. Since the filament pitch is increased in the case of a two-start helix, a two-start helix is more favorable compared to a one-start (Fig. 5d–e). Thus, our observation of two-start helices is consistent with the theoretically derived Stalk filament elastic energy. It should be noted that from a theoretical perspective it is not clear why three- or four-start helices have not also been observed, as their elastic energy would be even

lower. Perhaps the entropic cost is prohibitive, or the GG interactions are perturbed at the resulting higher pitch angles.

A further important question is whether a two-start geometry would preclude the filaments from generating further constriction through GTP-dependent sliding of the filament turns. We carried out simulation work based on the model of how a one-start helix evolves in the presence of GTP[23]. Starting with a two-start geometry, the result, similar to the one-start case, shows that the torques are non-canceling, and thus, the rungs can rotate and generate constriction (Supplementary Movie 1). Note that there is no geometrical constraint precluding two-start helices from forming helices with radii larger than the super-constricted state as is the case for the initial condition in Supplementary Movie 1.

## Discussion
Native dynamin has a long unstructured PRD domain at the C-terminus, which binds to the SH3 domain of partners to release its inhibition on forming helical assemblies[15,24,25]. To study an ordered helix in vitro, the PRD domain is truncated for most of the studies (Fig. 1a). In this study, even though an MBP domain was fused to the N-terminal of dynamin via a long linker, the linker and MBP domain occupy the equivalent physical space as the C-terminal PRD and the SH3 domain of partners would be. On the other hand, the low-resolution map of the two-start helix in the presence of GTP and lipid (EMD-7958) has a similar helical symmetry (rise 14.63 Å, twist 26.14°)[12]. The two-start helix studied in this report is lipid-free. Nevertheless, no significant difference has been observed between these two two-start helical maps (Fig. S6). Thereby, we believe our helix structure represents one of the conserved super-constricted two-start conformations.

The atomic structure of the two-start dynamin helix allows a detailed comparison of conformational differences with that of the one-start helix. Surprisingly, all the three Stalk interfaces are well conserved (Fig. S4), irrespective of helical symmetry. Furthermore, the Stalk interface 2 and interface 3 serve as fulcrum points to allow small relative rotations of the Stalk domains, while

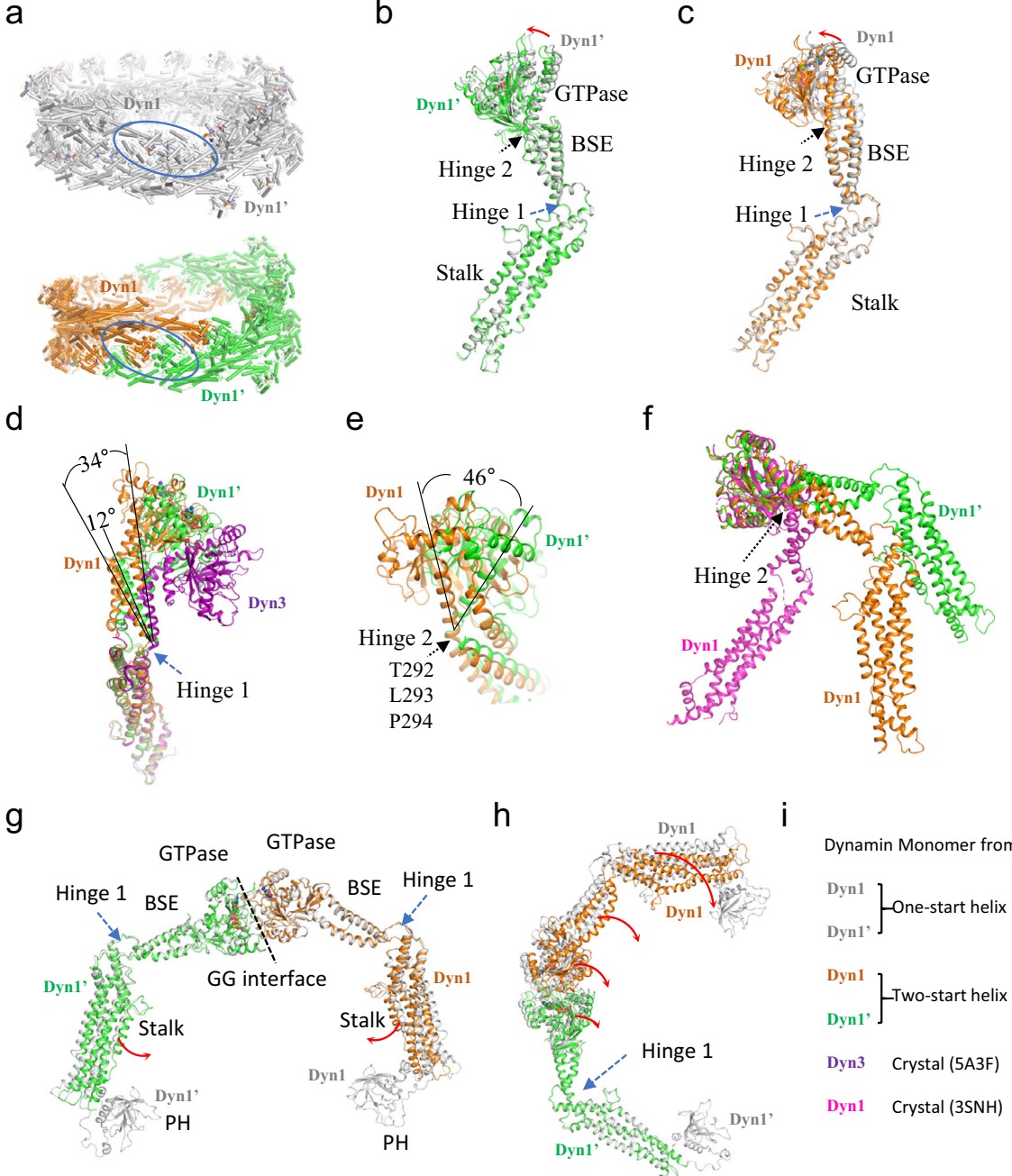

**Fig. 3 Analysis of the Hinge conformations. a** The GG dimers in the one-start and the two-start helix. The one-start helical strand is colored gray, and the two two-start helical strands are colored in green and orange, respectively. The GG dimers circled are used for structural analysis and monomers from them are colored accordingly through (**b**) to (**h**). **b, c** Structural comparisons of the two asymmetric monomers between the two helical symmetries. Monomers with similar Hinge 2 conformations between the one-start and the two-start helix are aligned based on the Stalk domains. 1.8° (**b**) and 4.2° (**c**) rotations around Hinde 1 are observed. **d** Flexibility of the Hinge 1 conformation. The Stalk domains are aligned between the two monomers from the two-start helix and the crystal structure (colored in purple, PDB: 5A3F). **e** Conformations of Hinge 2 of the two monomers from the two-start dynamin helix. One monomer (colored in orange) has a kink centered around T292 and P294, which in the other (colored in green) is a continuous helix. The GTPase domain swings 46° around Hinge 2 between the two monomers when the BSE domains are aligned. **f** Flexibility of Hinge 2 conformation. The GTPase domains are aligned between the two monomers from the two-start helix and the crystal structure (colored in magenta, PDB: 3SNH). **g, h** Hinge 1 alters the Stalk orientation within the GG dimer. The GTPase domains are aligned in (**g**) and the Stalk domains of the bottom molecule are aligned in (**h**). Red arrows indicate the domain movements, black dashed arrows indicate the Hinge 2 locations, the blue dashed arrows indicate the Hinge 1 locations, and black dashed lines indicate the GG dimeric interface. **i** Color code of the dynamin monomers.

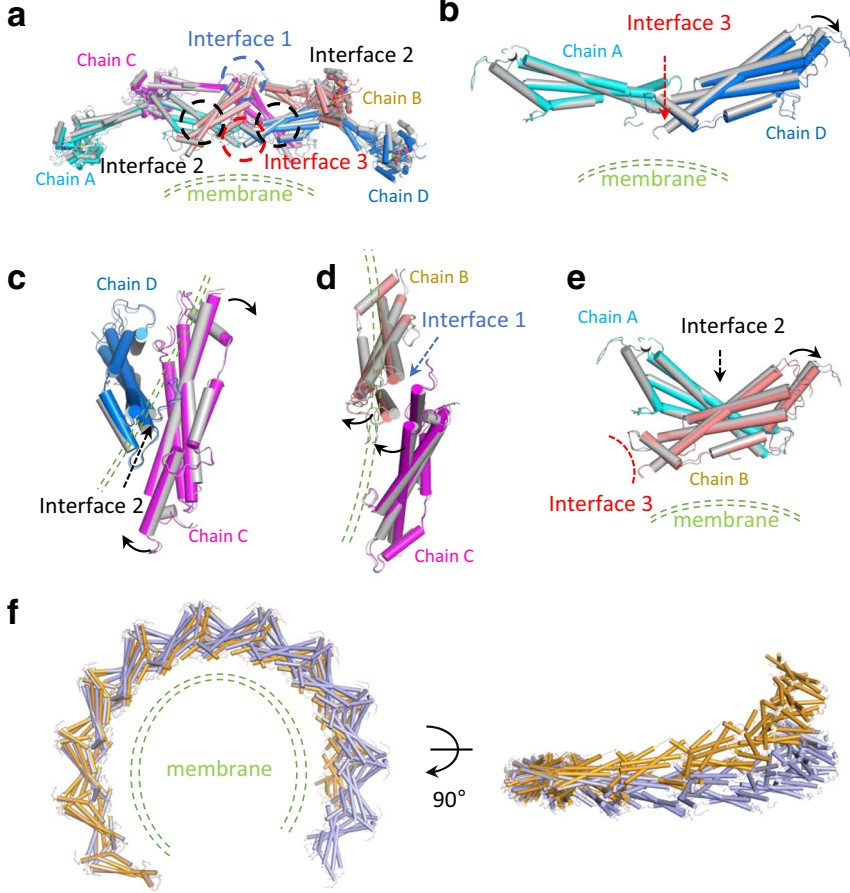

**Fig. 4 Comparison of the Stalk conformations between one-start and two-start Dyn1ΔPRD assemblies. a–e** Comparison of Dyn1ΔPRD Stalk tetramers from one-start and two-start assemblies (**a**), around the interface 3 (**b**, **e**), interface 2 (**c**), and interface 1 (**d**). The one-start tetramer is in gray and the two-start tetramer is colored. One Stalk domain (colored in cyan) from the two-start helix is aligned to its counterpart. The black arrows indicate the movements from one-start conformation to the two-start one. The detailed view of each interface is shown in Fig. S4. **f** The small motions within the Dyn1ΔPRD Stalk tetramer, which resulted in a small increase in the rise of individual assembly subunits (from 6.3 Å in one-start to 13.6 Å in two-start), propagates and accumulates through a helical turn. For clarity, only the Stalk domains of one rung are shown. The one-start is colored in light blue and the two-start is colored in orange. The putative location of the lipid bilayer is drawn in green dashed lines throughout the panels. Note, in panels (**c**) and (**d**), the membrane is located above and behind the page, respectively.

preserving interface 1 accordingly. The Hinge 1 conformational change between the one-start and two-start helices represents a mechanism to transduce the energies from the GTPase domain to the PH domain and vice versa. The asymmetry in Hinge 1 conformations appears to contribute to the conformational transmission. Meanwhile, the asymmetry in Hinge 2 conformations likely confirms the power-stroke-like mechanisms of the GTPase-BSE motor, where GTP hydrolysis swings BSE to twist and constrict the underlying membranes[26].

In simulations informed by experimental characterization of dynamin's GTPase cycle, motor activity acting between neighboring turns of a one-start helix has been shown to sufficiently constrict an underlying membrane to promote scission[23]. When dynamin initially oligomerizes on lipids, its filament's intrinsic curvature can already partially constrict lipids in a GTP-independent manner (Fig. 5e, blue arrow). Strong dimerization between turns, promoted by GTP, may counteract the preferential expansion of the pitch. During the GTP-dependent constriction, the GTPase cycle creates strong GG dimers and drives these dimers to shorten and drag neighboring turns relative to one another before dissociating and starting again. Due to these strong forces, the dynamin filament may be able to deform away from its elastically preferred configuration, leading to a rise in

elastic energy and a significant increase in elastic stress in the filament (Fig. 5e, orange arrow). If the connection between rungs, the GG dimers, can resist this stress, a one-start helix can catalyze membrane fission. Once the fission has been achieved, the lack of support of the underlying membrane tube template would stimulate the dynamin filament to disassemble. The accumulated stress in the super-constricted one-start filament would likely facilitate the disassembly, possibly through an extension of the pitch (Fig. 5f, upper).

On the other hand, there are a few reported cryoEM reconstructions of dynamin filaments (Fig. S7), and they could be coarsely divided into two groups: the constricted tubes featuring an inner lipid tube diameter of 7 nm and the super-constricted tubes featuring an inner lipid tube diameter 4 nm. Surprisingly, all the constricted reconstructions are in one-start helical configurations, and all the super-constricted reconstructions are in two-start helical configurations, irrespective of varying helical parameters in rise and twist (Fig. S7). Even though the lack of an observation of a one-start super-constricted helix in vitro could be explained by its intrinsic instability as described above (Fig. 5e), it emphasizes the possible significance of the two-start super-constricted helical conformations. Considering an inner lipid tube diameter of 4 nm is close to the theoretical limit of

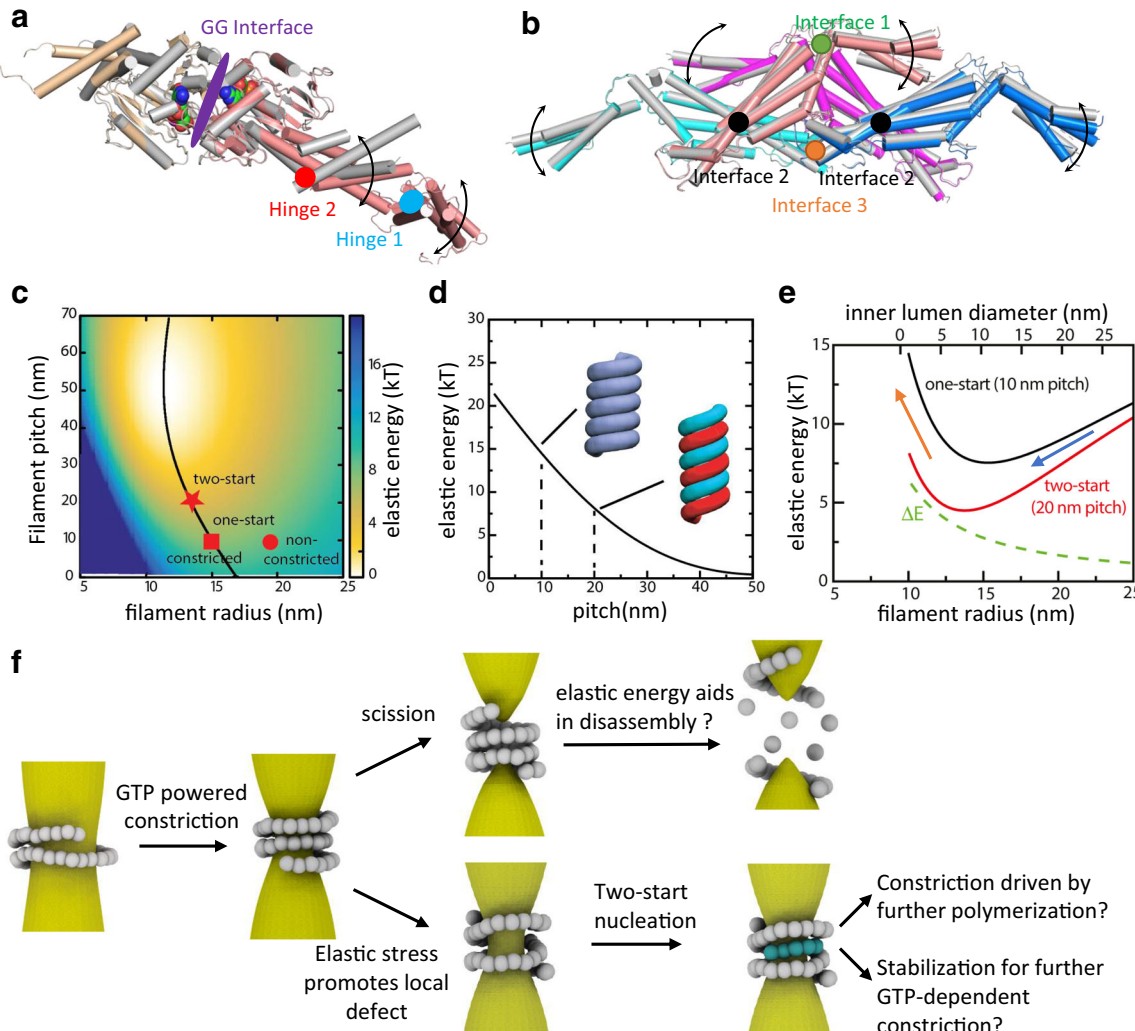

**Fig. 5 Possible roles for elastic energy during dynamin-mediated membrane fission. a** Schematic showing the conserved GG interface and the motions around the Hinge 1 and the Hinge 2. **b** Schematic showing the three conserved Stalk interfaces and the Stalk motions around them. **c** Elastic energy per dimer plotted over a range of Stalk filament radii and pitches. Minimal values are reached at large pitches of ~50 nm. Black line follows the minimal value of the elastic energy as a function of the filament pitch. Helical parameters for the super-constricted structures solved as two-start helices lie on the minimal energy line. **d** Elastic energy per dimer versus pitch for a constant filament radius of 13.5 nm. The elastic energy predicts that a two-start helix (cyan/red) is more favorable than a one-start helix (blue) because the two-start configuration increases the filament pitch without disrupting the stabilizing GG interactions between neighboring turns. Note that in (**c–e**) the filament radius is defined by the distance between the helical axis and interface 2. **e** The elastic energy per dimer is shown as a function of the filament radius (lower axis) or the inferred inner lumen diameter of the underlying membrane tube (upper axis). Constriction proceeds downhill in elastic energy (blue arrow) until the inner lumen diameter reaches ~10–12 nm for one-start (black line) or ~5–7 nm for two-start (red line). After this point, active constriction driven by GTP works against an uphill filament energy, and, therefore, filaments around tightly constricted membrane necks show an increase in elastic energy (orange arrow) in both one-start and two-start helices. Importantly, taking the difference between the one-start and two-start elastic energy shows that the two-start is relatively more stable as the filament constricts (green dotted line). **f** Working model of membrane fission. Dynamin initially polymerizes around a lipid tube. GTP-powered constriction results in a reduction of the membrane tube radius (left). In an actively constricting one-start filament, the pitch is constant while the radius is shrinking, leading to an increase in elastic stress within the filament. This stress build-up could possibly aid in the disassembly of the dynamin collar after the scission of the neck (top). Meanwhile, elastic stress could increase the chance of a local detachment between neighboring filament turns. Such structures provide the possible nucleation sites for forming two-start helices (bottom). Converting into an elastic energy favorable state through pitch extension could possibly aid further constriction of the membrane tube. An intercalating filament would add stability to the dynamin collar by forming new GG dimeric interface with the neighboring rung.

spontaneous membrane fission, it is, therefore, plausible to suggest that the two-start helical conformation of dynamin could be the conformational intermediate before the final membrane scission. The elasticity analysis indicates that the elastic energy strongly increases as the filament is tightly constricted and that a two-start geometry will significantly mitigate this increase (Fig. 5d, e). Since the GTPase cycle inherently involves the

transient dissociation of GG dimers, pitch-increasing elastic stress may lead to local defects that can be repaired by new dynamin tetramers joining in (Fig. 5f bottom). It could thus start further polymerizations featuring the extension of the pitch, the drop of the elastic stress, and the reduction of the diameter of the dynamin filament and underlying membrane. In this case, the stabilized filament may hold the membrane tube in the vicinity of

the super-constricted diameter allowing time for scission to occur. The smallest inner diameter of an in vitro assembled membrane-bound two-start dynamin filament in the presence of the nucleotide is around 3.4 nm (Fig. S7e), which could lead to membrane fission spontaneously. An in vitro reconstituted system showed an extension of dynamin helical pitch upon the nucleotide hydrolysis when using rigid lipid tubes as the membrane template[27]. Light microscopy analysis showed twisting and supercoiling of dynamin tubes upon addition of GTP, suggesting a relative rotation of the helix turns during GTP hydrolysis[28]. Both are likely to support this conversion model.

Alternatively, GTPase activity could continue to play an active role within a two-start geometry. The simulation results show that the two-start helices are able to constrict the underlying membrane in a GTP-powered manner (Supplementary Movie 1). Even though the 3D reconstructions of the two-start helices are all in the super-constricted state, it is possible that dynamin molecules could wrap around membrane templates with wider diameters (>4 nm) in two-start-like conformations (large pitch), which would then employ active constriction to drive scission.

In summary, it is clear that dynamin constricts the lipid using the energy of GTP hydrolysis, and it is likely that both one-start and two-start helices are involved (Fig. 5f). The small rearrangements between dynamin monomers lead to different helical symmetries, which contribute to membrane fission. Both one-start and two-start helices could lead to scission theoretically, and further investigations are required to differentiate their cellular impacts.

## Methods

**Plasmid construction.** The cloning vector containing the gene for the wild-type human dynamin 1 lacking the PRD (UniProt ID: Q05193) was a gift from Katja Fäelber (Structure and Membrane Interaction of G-Proteins, Max-Delbrück-Centrum für Molekulare Medizin) and that for the MBP tag was a gift from Jinwoo Ahn (Pittsburgh Center for HIV Protein Interactions, University of Pittsburgh School of Medicine). The expression vector pcDNA3.1(+) was obtained from Life Technologies (Invitrogen). The dynamin 1 gene and the MBP tag were amplified by polymerase chain reaction (PCR) (Supplementary Table 2) and then subcloned, using the NEBuilder HiFi Assembly kit (New England Biolabs Inc.), into pcDNA3.1(+) that had been linearized by the restriction enzymes Eco RV and Xba I. The resulting insert, designated as MBP-Dyn1ΔPRD, has a leading Kozak sequence, an N-terminal MBP tag, a hexahistidine tag, followed by a human rhinovirus 3 C protease cut site, and the wild-type Dyn1 ΔPRD. Mutations on this construct were introduced by site-directed mutagenesis using overlapping primers. The sequences of the inserts and mutations were confirmed by DNA sequencing (Genewiz Inc.).

**Expression and purification of MBP-Dyn1ΔPRD.** Recombinant MBP-Dyn1ΔPRD was transiently expressed in mammalian cells using the Expi293 Expression kit from Life Technologies (Invitrogen). Suspension-adapted Expi293F cells were grown in Expi293 Expression Medium to a density of $3.5 \times 10^6$ to $4 \times 10^6$ cells/ml and viability of >95% 24 h before transfection. Plasmid DNA and Expi-Fectamine reagent (Invitrogen) were diluted in Opti-MEM I Reduced Serum (Invitrogen) into separate tubes, incubated for 5 min at room temperature, and then mixed together for 25 min. Cells were transfected with the DNA-ExpiFectamine complex at a DNA/transfection reagent/cell culture volume ratio of 30 μg/1.5 ml/30 ml and to a final cell density of $2.9 \times 10^6$ cells/ml. Cells were then incubated at 37 °C and 125-rpm agitation with 8% CO2 in the air. After 18–24 h of incubation, instead of adding the enhancers as per the manufacturer's protocol, the cells were immediately harvested by centrifugation at low speed (100x$g$). Cells were then washed once with cold phosphate-buffered saline, and the cell pellet was flash-frozen and stored at −80 °C for later use.

The thawed cell pellet was resuspended in buffer A [50 mM Hepes-KOH (pH 8), 500 mM NaCl, and 5% glycerol] supplemented with detergents (1% Tween 20 and 0.3% NP-40), deoxyribonuclease I (50 μg/ml; Sigma-Aldrich) in the presence of 5 mM MgCl₂, 10 mM β-mercaptoethanol, and a cocktail of protease inhibitors (Roche). After 1 h of rotation at 4 °C, the lysate was homogenized by 15 strokes in an ice-cold, tight-fitting Dounce homogenizer. The homogenate was then centrifuged at 21,000x$g$ at 4 °C for 30 min. After centrifugation, the supernatant was collected and mixed with 1 ml of amylose agarose resin (New England Biolabs Inc.) (per 50 ml of cell suspension) pre-equilibrated with buffer A. The mixture was incubated with rotation at 4 °C for 1 h and then transferred to a column to flow-through. The resin was washed with 50× resin volume of buffer A. For some of the mutants, the NaCl concentration was increased to 750 mM NaCl during the wash before continuing with buffer A. To elute the recombinant protein, the resin was incubated, in batch, with buffer A containing 50 mM maltose for 15 min at 4 °C, and then the flow-through was collected as elution. The purified protein was detected by Western blot using an antibody against dynamin, anti-Dynamin 1 antibody (500x dilution) (ab14448, Abcam).

**Preparation of liposomes.** Synthetic 1, 2-dioleoyl-*sn*-glycero-3-phosphoserine (DOPS) lipid in chloroform (Avanti) was dried under a stream of high purity nitrogen and left in a vacuum desiccator for at least 1 h. The lipids were rehydrated in 20 mM Hepes-KOH (pH 7) and the liposomes were prepared by extrusion through 1-μm-pore polycarbonate membranes (Whatman) using an Avanti Mini-extruder.

**GTPase assay.** The GTPase activity of MBP-Dyn1ΔPRD, with or without DOPS liposomes, was assessed using a continuous NADH (reduced form of nicotinamide adenine dinucleotide)–coupled assay[29]. The reaction mixture was prepared to achieve the following final concentrations in the assay solution: 50 mM Hepes-KOH (pH 7), 150 mM NaCl, 4 mM MgCl₂, 4 mM phosphoenolpyruvate (PEP), 2 mM NADH, and 10 U of pyruvate kinase/lactate dehydrogenase. Dynamin (1 μM) was mixed with 160 μM liposomes or 20 mM Hepes-KOH (pH 7) and the reaction was initiated with the addition of 1 mM GTP. The decrease in the NADH absorbance at 340 nm was monitored in a 96-well plate using a Tecan SPARK 20 M (Tecan) at 30 °C. The rate of NADH oxidation was measured and used to calculate the $k_{obs}$ of MBP-Dyn1ΔPRD. NADH oxidation was also monitored in the absence of protein using buffer only as control. The experimental values were normalized to correct for background. Results are representative of three independent measurements.

### Electron microscopy

*Sample preparation.* Helical assembly of MBP-Dyn1ΔPRD was performed in assembly buffer [20 mM Hepes-KOH (pH 7), 150 mM NaCl, 1 mM MgCl₂, and 2 mM EGTA] at 0.5 mg/ml dynamin concentration, 1 mM GMP-PCP and/or 0.5 mg/ml DOPS liposomes where indicated.

*Negative stain EM.* Aliquots (3 μl) from the helical samples were adsorbed to a glow-discharged, 400-mesh, carbon-coated copper grid and stained with fresh uranyl formate (2%). Images were recorded on a TF20 electron microscope (FEI) equipped with a field-emission gun at the indicated magnification on a 4k × 4k Gatan UltraScan charge-coupled device camera (Gatan).

*Immunogold labeling.* Immunogold labeling was performed to determine the location of the MBP in the helical assembly of dynamin. Samples containing the dynamin tubes were prepared and applied to a grid, as described above. The grid was successively floated on the following solutions: (i) twice with blocking buffer [bovine serum albumin (BSA;10 mg/ml) in oligomerization buffer] for 5 min, (ii) with blocking buffer containing primary antibody against MBP tag (Abcam) (1:250 dilution) for 1 h, (iii) twice with blocking buffer for 5 min, and (iv) with blocking buffer containing a 5-mm gold-labeled secondary antibody (Ted Pella) (1:250 dilution) for 1 h. All incubations were carried out at 4 °C, and the grid was washed once with blocking buffer and twice with oligomerization buffer before staining with uranyl formate.

*CryoEM.* Three microliters of the dynamin tubes (0.5 mg/ml) was applied on the carbon side of glow-discharged holey R2/1 Quantifoil grids (Quantifoil Micro Tools GmbH), manually blotted from the backside, and then plunge-frozen in liquid ethane using a homemade manual plunger. The grids were imaged on a Titan Krios microscope (Thermofisher Scientific, USA) operated at 300 KeV, with a nominal magnification of 130,000 × in EFTEM mode on a post-column Quantum-K2 detector (Gatan, USA) operated in counting mode with an energy selecting slit of 20 eV. Detailed parameters for data collection are listed in Table 1. All data were collected using EPU software (Thermofisher Scientific, USA).

**Image processing and helical reconstruction.** Pre-processing of the movies from both data collection sessions were managed using Scipion[30], such that the movies were imported and then motion-corrected and dose compensated using MotionCor2[31]. The dose compensated aligned sums were then imported into RELION 2.0 and RELION 3.0[32,33] for all further processing was done (Table 1). CTF estimation was done with CTFFIND4[34] and micrographs with poor estimated CTF solutions were discarded. The helical tubes were then manually picked and segments were extracted with an inter-box distance of 68 Å. Multiple rounds of 2D and 3D classification are carried out. A featureless cylinder with a diameter of 380 Å was used as the initial reference for 3D classification with helical parameters estimated by indexing of diffraction patterns calculated from two-dimensional class averages and single tubes. The reconstructions showing the best structural detail after 3D classification were chosen for further refinement (Table 1). The refined segments from

NT14393-32 and NT14393-34 were then combined and another round of classifications was performed. Per particle, CTF values were optimized[35]. The final refinement is carried out in RELION 3.0 auto-refine, with helical (rise 13.58 Å and twist 24.43°) and C2 symmetries applied. The FSC and B factor for the combined data were then calculated by RELION 3.0 postprocess (Table 1). Chimera 1.15 (https://www.cgl.ucsf.edu/chimera/) and ChimeraX 1.1 (https://www.rbvi.ucsf.edu/chimerax/) are used for the visualization of electron density maps.

**Model building and refinement**. The two monomers of dynamin dimeric structure from one-start helix (PDB: 6DLU)[12] are fitted into the electron density in coot[36]. Iterative rounds of manual adjustment in coot 0.9.4.1 and real-space refinement in Phenix 1.18.2–3874[37] are carried out. The resulting model is used to generate a full helix model, which is further refined by Phenix 1.18.2–3874. The final refinement statistics are shown in Table 1.

**Theoretical analysis of Stalk filament elasticity**. The dynamin Stalk dimer is the building block of the Stalk filament and the connections between Stalk dimers are defined by interfaces 1 and 3 of the Stalk tetramer. A previous study[22] used molecular dynamics simulations of the dynamin Stalk tetramer to extract the spontaneous curvatures and elastic moduli of the dynamin filament. Here, we use these parameters to write down the elastic energy per unit length, $E$, of the dynamin Stalk filament:

$$E = \alpha_\kappa (\kappa - \kappa_0)^2 + \alpha_\tau (\tau - \tau_0)^2 \qquad (1)$$

where $\alpha_\kappa$ and $\alpha_\tau$ are the elastic moduli, $\kappa_0$ and $\tau_0$ are the spontaneous curvatures, and $\kappa$ is the filament curvature and $\tau$ is the twist. The parameter values are $\alpha_\kappa = 3000$ nm•kJ/mol, $\alpha_\tau = 2700$ nm•kJ/mol, $\kappa_0 = 0.058$ nm$^{-1}$, and $\tau_0 = 0.041$ nm$^{-1}$. In general $\kappa$ and $\tau$ can vary along the filament, however, here, we limit our treatment to the long and constant helices observed by structural biology, and, thus, $\kappa$ and $\tau$ are constant within a particular helical arrangement. The plotted energy is energy per dimer, i.e., $E \times (5.6$ nm$)$. In order to plot in terms of the more intuitive measures of the radius $r$ and pitch $p$, the following relations were used:

$$\kappa = \frac{r}{h^2 + r^2} \qquad (2)$$

$$\tau = \frac{h}{h^2 + r^2} \qquad (3)$$

where $h = p/2\pi$. When the filament forms GG dimers between its turns this contributes an additional free energy per unit length of $g_0$, and the total energy per unit length becomes:

$$E_{total} = E + g_0 \qquad (4)$$

Since we have shown that the GG interface and, thus, $g_0$ is the same between one and two-start helices, $g_0$ can be neglected in the comparisons. Figure 5c plots $E(r,p)$ for a range of radii and pitches. Figure 5d plots $E(r = 13.5$ nm$, p)$ since 13.5 nm is the radius of the two-start structure. Figure 5e plots $E(r,p = 10$ nm$)$ since 10 nm is the pitch of a one-start helix. The filament radius is defined by the distance between the helical axis and interface 2.

**Simulation model for two-start constriction dynamics**. The proof-of-concept simulation for two-start constriction was generated using the recent coarse-grained active (GTP-dependent) constriction model for dynamin[23]. Refer to that study for the details of the model interactions, and all simulation parameters are the same. The code used to implement the constriction model is publicly available at https://bitbucket.org/jknoel/constrictionsimulation. In brief, the dynamin filament was represented as an elastic polymer with stiffness parameters described above and with each bead corresponding to a dynamin dimer. The membrane tube description was based on an axially symmetric continuous Helfrich elastic membrane with stiffness $\chi$ and under tension $\gamma$. The membrane stiffness $\chi = 24$ $k_BT$ and the tension $\gamma = 0.03$ $k_BT/nm^2$ were chosen. For Supplementary Movie 1, the filaments are initially wrapped around a membrane at its equilibrium diameter (40 nm) and each with a pitch of 20 nm. Then, in the presence of 300 μM GTP, the system actively constricts to an inner diameter of 8 nm in 300 ms.

**Reporting Summary**. Further information on research design is available in the Nature Research Reporting Summary linked to this article.

## Data availability

The CryoEM density map of Dyn1ΔPRD has been deposited in the EM Data Bank (EMDB) under the accession code EMD-11932. The atomic model of the Dyn1ΔPRD helix has been deposited in the Protein Data Bank (PDB) under the accession code 7AX3. Uncropped blots and gel images are supplied as the Supplementary Fig. 8.

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

## Acknowledgements

We thank C. Xu, J. R. Perilla, and J. Ning for scientific discussion and technical support, J. Dong, R. Esnouf, and C. Freeman for computer system support. We acknowledge Diamond Light Source for access and support of the CryoEM facilities at the UK national electron bio-imaging center (eBIC), proposal NT14393, funded by the Wellcome Trust, MRC, and BBSRC, and thank eBIC staff Dr. Yuriy Chaban for help with data collection. This work was supported by the National Institutes of Health (P50AI150481, P.Z.), the UK Wellcome Trust Investigator Award 206422/Z/17/Z (P.Z.), and the UK Biotechnology and Biological Sciences Research Council grant BB/S003339/1 (P.Z.).

## Author contributions

P.Z. conceived the project. F.J.D.A. purified and characterized the proteins and made the cryoEM sample. D.K.C. collected cryoEM data. J.L., D.K.C., and F.J.D.A performed the reconstructions. J.L. built the model. J.K.N. carried out an elastic analysis. J.L. and J.K.N. developed the working model with contributions from P.Z. J.L. and P.Z. analyzed the data and wrote the paper with contributions from all authors.

## Competing interests

The authors declare no competing interests.
