## [Peer Review File · Nature Communications]

CryoEM structure of the super-constricted two-start dynamin 1 filamentREVIEWER COMMENTS

Reviewer #1 (Remarks to the Author):

Dynamin is the prototypical member of a superfamily of multidomain GTPases and catalyzes membrane scission at the plasma membrane. This activity requires dynamin self-assembly into helical polymers, which in turn promotes G domain dimerization and stimulated GTP hydrolysis. The subsequent structural changes that occur throughout the polymer and ultimately lead to fission, however, remain poorly defined and hotly debated. Previous studies yielded structural snapshots of dynamin polymers with and without lipid templates at varying resolutions. These include a constricted state with one-start helical symmetry and a super-constricted state with two-start symmetry whose geometry approaches the theoretical limit for spontaneous hemi-fission. Understanding the transitions between these states is important for unraveling the molecular mechanisms governing dynamin's actions in membrane remodeling. Here Liu et al. describe the 3.74-Å cryo-EM structure of a two-start helical polymer of MBP-tagged Δ PRD dynamin 1 assembled in the absence of lipid. The structure shows that although the G domain dimer interface between subunits remains largely unchanged throughout the assembly, the orientation of the BSEs around hinge 2 is asymmetric. Structural comparison with the constricted one-start assembly reveals changes in the stalk orientation relative to hinge 1 and that small changes within the tetramer propagate throughout the polymer to radically alter the helical pitch. From these observations, the authors argue in support of a constriction/ratchet model. While the structure presented here represents a significant improvement in terms of resolution compared to earlier reconstructions of two-start dynamin helices (Sundborger et al., 2014; Kong et al., 2018), the manuscript fails to leverage this new atomic-resolution information in any meaningful way and essentially describes a series of superpositions looking at domain movements and interface fluctuations with little attention paid to the underlying side chains. A dearth of any biochemical, biophysical, or cell biological data further testing the mechanistic insights gleaned from the authors' structural comparisons diminishes the overall impact of this work. As such, this manuscript would be better suited to a more specialized journal.

Comments:

1) The introduction seems a bit narrow in scope would be improved with a number of changes and additions.

Page 3, second paragraph:

It should be explained that in solution the dynamin tetramer exhibits basal GTP hydrolysis and that dynamin assembly into helices leads to stimulated GTP hydrolysis by promoting the G domain dimerization between tetramer subunits in adjacent helical rungs. The current description as written ("Dynamin exhibits assembly stimulated GTPase activity...") is vague and doesn't state where and how dimerization occurs in the context of the assembly or that it doesn't occur in the context of the tetramer. The statement "Dimerization between GTPase domains is subject to the nucleotide states" is similarly ambiguous and should be clarified.

Page 3, third paragraph:

A description of and comparison with the GTP-bound dynamin 1 reconstruction (Kong et al., 2018) should be included along with the GTP-bound K44A structure as it too forms a two-start helix that achieves a 'super-constricted' diameter and preceded the present study. It should also be noted somewhere that two-start assemblies have also been observed Dnm1 even with GMPPCP (Mears et al., 2011).

Page 4, first paragraph:

It would benefit the reader to explain how the super-constricted state relates to hemi-fission and why achieving such a narrow diameter is thought to be biologically and biophysically significant (see Bashkirov et al., 2008; Sundborger et al., 2014). Providing this connection will underscore why a two-

start symmetry is significant.

Adding a paragraph to the introduction that summarizes the current models for fission would help set the stage for the findings described in this work, especially since they are brought up again in the discussion.

2) The second paragraph of the introduction enumerates all the domains, hinges, and interfaces that form the building blocks of the dynamin polymer and play an important role in structural transitions. It would be helpful to include a supplementary figure that visually summarizes all these important features in place for unacquainted readers and less structurally inclined cell biologists.

3) Page 4: "We collected cryoEM of these dynamin tubes": It should be explicitly stated here that your reconstruction was carried out on MBP-tagged Δ PRD polymers in the absence of lipid. It is vague and confusing as written, as the previous sentence and figure legend references (Fig. 1f) refer to lipid bound tubes labeled with immuno-gold antibodies.

4) Fig. 3: Inclusion of the one-start comparisons in panels 3C and 3D here seems out of place and would fit better with Fig. 4, especially since it referenced in the text when talking about the hinge 1 movements.

5) Page 6, first paragraph: "In one start helix, the GG dimer interface is formed by the GTPase domains of the same strand". While this statement is technically true because it's only one single strand wrapping around, it's vague as written as it doesn't distinguish between dimers forming between adjacent helical rungs of the same strand versus adjacent neighboring subunits within the strand. It should be clarified to state that dimerization in a one-start helix only occurs between tetramers in the same strand following the completion of a helical turn. Additionally, either "helix" should be changed to "helices" or an "a" should be added to read "In a one start helix..."

6) Discussion: It would be beneficial to expound upon the different models for dynamin catalyzed fission in the discussion and provide a more detailed description for exactly how the observations presented here add to the interpretation. A more direct comparison to the model suggested by Kong et al., 2018 describing how hydrolysis-dependent changes in the BSE and stalk could help drive constriction would be particularly warranted, especially since this incorporated a similar symmetry transition based on a lower-resolution two-start assembly.

Reviewer #2 (Remarks to the Author):

Dynamin-1 is a long-studied and fascinating GTPase involved in membrane fission reactions. Liu et al. report the $\sim 3.7\text{\AA}$ reconstruction a 2-start filament of polymeric dynamin-1. The structure is remarkably similar to the 1-start filament structure published in 2018, including the up/down asymmetry in hinge-2 centered near Ile289 and Arg290. Comparing the two morphologies shows us how very small rotations in the stalks, when summed in serial along the filament axis, lead to a dramatic change in the pitch of the helical assembly. This structural insight should be published-- pending minor improvements. However, we recommend re-writing certain sections to be more cautious about the biological meaning of the 2-start helix. The functional model for fission via a 2-start structure remains unsubstantiated. The sample, moreover, has liabilities that limit insight into membrane fission mechanisms.

Recommendation:

To publish with softer functional claims, a broader discussion of the literature, and acknowledgment of the sample's limitations.

Major questions:

1. The paper is structure-only, without validating mutants tested in vitro or in cell experiments. Therefore, we advocate making the structural model as reliable as possible. The Rama-Z score provided by current versions of phenix reports a standard score for the overall Ramachandran space: the distribution of the (ϕ , Ψ) torsion angles of the protein backbone. We mostly strive for "zero unexplained outliers", where explained outliers are justified by high-quality density. However, in the $\sim 3.5\text{\AA}$ -to- 4\AA resolution regimen, where the density is usually not sufficient to justify outliers, one can overfit the model's ϕ and Ψ torsion angles. The Rama-Z quality metric measures the distance from the mean distribution to the observed distribution (how many standard deviations below or above the population mean a raw score is). This 2-start model falls in the Rama-Z "suspicious" region, suggesting that refinement software forced outlier residues into specific regions of the allowed, but not necessarily correct, dihedral space. We recommend the author's evaluate whether the model could be further improved before depositing the structures.

2. Related to #1, the clash score is high. We have seen this as well with filamentous samples, where clashes at the interface between subunits are common and challenging to address with current refinement algorithms. We hope further attention to model building will bring the clashscore down, but please still deposit the biological assembly as a helix (as you shared) rather than reduce the model to just a monomer or dimer (which would lead to better statistics, but a less useful pdb deposition for the community).

3. Hinge 2 and its asymmetry is a highlight of the paper. Please consider expanding the discussion of hinge 2 to include comparison of the 2-start with the 1-start reported by Kong et al. In the 2-start, this hinge appears to be a kink in the helix, rather than a loop between two helices? How would the authors interpret the Kong et al. observations regarding mutations in this region? Specifically, they report that a triple mutant that may boost the helical propensity near the Pro side of the kink (T292A/L293A/P294A) did result in a transferrin uptake deficiency--but the single mutant Pro294=>A, and a triple mutant closer to the residues highlighted here (R290A/D291A/T292A), was also fine for transferrin uptake?

Minor questions:

1. Include helical parameters used in table 1

2. Line 81 - please mention nucleotide state

3. Line 139-140- If hinge 1 is so flexible, by what atomic mechanism would it communicate conformational change? Please hypothesize with data from the structure or remove this sentence.

4. Fig S3. is busy and unlabeled. The message is difficult to appreciate.

5. Fig 5. It is difficult to keep track of the rotations.

Please consider illustrating where the membrane would be and the orientation of a larger helix with these views: a) membrane down, b, membrane down, but rotated, c) membrane behind page and d) membrane behind page, but rotated.

6. Please include a map-to-model FSC plot and filter the map to the appropriate resolution. Table suggests the model resolution is better than the half map resolution, which is unusual and hard to believe.

7. Consider expanding the discussion to acknowledge that this sample has liabilities that limit our ability to draw functional conclusions. These liabilities include i) the absence of a membrane and unconstrained PH domains, 2) a large MBP fusion on the N-terminus and 3) a deletion of the C-

terminus.

Point by point responses to reviewers:

Reviewer #1 (Remarks to the Author):

Dynamin is the prototypical member of a superfamily of multidomain GTPases and catalyzes membrane scission at the plasma membrane. This activity requires dynamin self-assembly into helical polymers, which in turn promotes G domain dimerization and stimulated GTP hydrolysis. The subsequent structural changes that occur throughout the polymer and ultimately lead to fission, however, remain poorly defined and hotly debated. Previous studies yielded structural snapshots of dynamin polymers with and without lipid templates at varying resolutions. These include a constricted state with one-start helical symmetry and a super-constricted state with two-start symmetry whose geometry approaches the theoretical limit for spontaneous hemi-fission. Understanding the transitions between these states is important for unraveling the molecular mechanisms governing dynamin's actions in membrane remodeling. Here Liu et al. describe the 3.74-Å cryo-EM structure of a two-start helical polymer of MBP-tagged ΔPRD dynamin 1 assembled in the absence of lipid. The structure shows that although the G domain dimer interface between subunits remains largely unchanged throughout the assembly, the orientation of the BSEs around hinge 2 is asymmetric. Structural comparison with the constricted one-start assembly reveals changes in the stalk orientation relative to hinge 1 and that small changes within the tetramer propagate throughout the polymer to radically alter the helical pitch. From these observations, the authors argue in support of a constriction/ratchet model. While the structure presented here represents a significant improvement in terms of resolution compared to earlier reconstructions of two-start dynamin helices (Sundborger et al., 2014; Kong et al., 2018), the manuscript fails to leverage this new atomic-resolution information in any meaningful way and essentially describes a series of superpositions looking at domain movements and interface fluctuations with little attention paid to the underlying side chains. A dearth of any biochemical, biophysical, or cell biological data further testing the mechanistic insights gleaned from the authors' structural comparisons diminishes the overall impact of this work. As such, this manuscript would be better suited to a more specialized journal.

We appreciate the reviewer for a nice overview of the topic and a summary of the data, and the positive remark that our structure “represents a significant improvement in terms of resolution”. It is indeed the structural changes throughout the polymer that ultimately lead to membrane scission. As the reviewer mentioned, it is thereby important to understand the different dynamin conformations and the possible transitions between them. Hence, by incorporating the constructive advices from the two reviewers, by additional theoretical calculations of elastic energies of different dynamin filaments and analyses of a wealth of existing mutagenesis data in the context of our structure, and by expanding our descriptions and discussions, we believe that the revised manuscript have provided significant improvements in our understanding of the dynamin mediated membrane fission. Details are described below.

First, for the reviewer's critique that “the manuscript fails to leverage this new atomic-resolution information in any meaningful way and essentially describes a series of superpositions looking at domain movements and interface fluctuations with little attention paid to the underlying side chains”, we apologize that the submitted manuscript was a short and concise version, such that we did not fully leverage this new atomic-resolution information and include the detailed descriptions and discussions of the underlying side chains that are relevant to the conclusion in this paper. We have now expanded the manuscript considerably with substantial details and made significant improvement by including additional figures, results and comprehensive discussions with relevance to current structure models and mutational and functional data. In particular,

1) we have modified and expanded our analysis of the Stalk interfaces in additional new Fig. S4 and detailed description on Page 7-8. We illustrated the side chains that constitute three conserved

Stalk interfaces of both the one-start and the two-start helices. We added a new supplementary table (Supplementary Table 1) to summarize previous mutagenesis data on these interface residues that support the functional essence and the conservation of these Stalk interfaces. A further comparison at residue level between the Stalk interfaces of the one-start and that of two-start helical assemblies, and between the two-start helical assembly and that of the “solution-state” crystal structure are also included in new Fig. S5.

2) in the new Fig. 3, we labelled the residues that constitute the Hinge 2 and also added descriptions of these residues and their role in mediating the conformational changes (Page 6). Since the densities at Hinge 1 could only allow the register of main chains, we did not mark specific residues that constitute the Hinge 1.

3) it is necessary to mention that the conformational differences between the protomers within the tetramer of different helical symmetries are tiny (Fig. 4), and it only becomes possible with a near-atomic resolution map to discern these small but significant conformation changes. Previous low-resolution maps led to tentative assumptions that the Stalk interfaces may alter between the different helical symmetries.

Second, for the critique that “A dearth of any biochemical, biophysical, or cell biological data further testing the mechanistic insights gleaned from the authors’ structural comparisons diminishes the overall impact of this work. As such, this manuscript would be better suited to a more specialized journal”, we understand that the reviewer is possibly looking for: 1) mutational analysis to validate the structure and the model, 2) utilizing other methods to further test the mechanistic insights derived from structural analysis, 3) as a result, what impacts could this paper have. Our responses are detailed below:

1) For structural validation by mutagenesis, there are already a wealth of previous mutational studies supporting the structure and interfaces, for example the Stalk interfaces shown in Fig S4, S5 and Supplementary Table 1. The GG and the Stalk interfaces are the keystones for the formation of the dynamin polymers, these interfaces are highly conserved as shown by our structural analysis (Fig. S3-S5, Fig. 3-4). Mutating interfacial residues resulted in deficiencies in the GTPase activity *in vitro*, the assembly ability *in vitro* or the transferrin uptake activity *in vivo*, irrespective of helical symmetries (Supplementary Table 1). For the Stalk interfaces, new panels are added in Fig. S4 and Fig. S5, and additional discussions are included (Page7-8). For the mutational analysis of Hinge 2, we added discussions on Page 7. All these mutagenesis results support and validate our two-start structure, the previously reported one-start structure and the crystal structures. It is worth noting that the interpretations of specific mutants should be placed in the context of different helical configurations, which become possible now with the our atomic two-start structure.

We understand there is a need to distinguish different dynamin helical assemblies, at least between one-start and two-start, experimentally by biochemical or biophysical methods. Unfortunately, it likely falls out of the capacity of many biophysical techniques, including single molecule imaging. Solving the structure by cryoEM is the best way or perhaps currently the only way to define the helical conformations of a dynamin polymer. At the moment, researchers in this field mainly take advantage of *in vitro* assemblies of different dynamin helical conformations to infer what is happening inside the cell. Then, it comes to the question on how to functionally differentiate one-start and two-start dynamin helices. Would it be possible to make mutations to test? Unluckily, the monomeric, the dimeric and the tetrameric dynamin are so similar between the one-start and the two-start helices (Fig. 3-4, Fig. S3-S4), that it becomes nearly impossible to rationally design a specific mutation(s) that would favour a specific helical conformation for functional analysis. One might screen a vast number of mutations through random mutagenesis to search for such mutants, but this is beyond the scope of this study.

2) Nonetheless, our structure provides an experimental benchmark for theoretical studies focusing on understanding the role of elastic energy in determining dynamin filament geometry. We thus elaborate on this topic through new theoretical work with our collaborator Jeffrey Noel from

MDC Berlin, who is an expert in elasticity theory and the theoretical treatment of dynamin-related proteins^{1,2}.

Fig. 5c-e plots the theoretically-determined elastic energies of the dynamin filaments. The elastic energy prediction was based on the molecular dynamics simulation-derived flexibility of a single dynamin stalk tetramer, isolating the filament energy from the influence of the membrane. Thus, our new dynamin structure, which shows the helical arrangement in the absence of membrane, offers a proper test case for the theory. The agreement between experiment and theory gives confidence in the elasticity results and, therefore, we proceeded to suggest two potential roles for the elasticity to play during dynamin mediated membrane fission (Fig. 5f).

3) This combination of experimental/theoretical understanding of the dynamin helical structure will have a significant impact on the field. Surprisingly, the dynamin filament is biased toward taking a larger pitch (Fig. 5c-d, lower energy), providing a rationale for the diversity of dynamin helical symmetries observed from us and other labs^{3,4}. The fact that significant elastic stress is generated within constricted one-start helices (Fig. 5e) may have functional consequences. We now discuss some possible mechanisms of dynamin-mediated membrane scission, as presented in Fig. 5f. When a one-start dynamin helices continuously and actively constrict the underlying lipid to a very narrow diameter, the dynamin filament itself is under stress (Fig. 5e). A possible escape to release the stress, but at the same time to maintain the constriction, is to locally nucleate a two-start geometry (Fig. 5f bottom). The dissociation of GG dimers is required to make room for the additional strand are readily generated from active GTP hydrolysis. An additional rung would enhance the stability of dynamin coat.

There will require more dynamin atomic structures at the different conformational states, or even time-resolved conformational switch in the future to provide the full conformational space that a dynamin filament could adopt, and theoretical simulations possibly plus other techniques such as atomic force microscope may provide the ultimate picture of the fission process. We envision it would take many years from the hard work of many brilliant labs. What we provided here is the first atomic structure of the two-start dynamin helix and the theoretical simulation data that rationalize and unifies almost all current structural results. We begin to place a few fundamental principles: 1) the Stalk interfaces are well-conserved irrespective of helical symmetries, and 2) dynamin intrinsically and energetically favours a large pitch (it is the two-start in comparison with the one-start helical configurations in this study). Therefore, we believe the impact is vital, as we provide not only the first atomic structure of a two-start super-constricted dynamin filament, but also insights into how different dynamin filaments may transit to one another.

In summary, by analysing existing mutational data in the context of our new structure, employing new theoretical computations, and expanding our text and figures for clearer descriptions, we believe we have significantly improved our manuscript and provided the potential for broad impact. We have 1) validated our structure by re-interpreting existing mutagenesis data, 2) highlighted the conservation of Stalk interfaces at atomic level which disfavoured the assumptions that Stalk interfaces in dynamin filaments vary with different helical geometries, 3) rationalized and unified almost all existing dynamin conformations and provided the potential functional and mechanistic significances of the two-start helices through theoretical simulations. These results will be of broad interests to researchers in the field of structure biology, biochemistry and cell biology of endocytosis and other membrane remodelling systems, *in-silicon* simulations of membrane remodelling events, the pathogenesis of dynamin related diseases. We hope you to take them into considerations when assessing our manuscript.

Comments:

1) The introduction seems a bit narrow in scope would be improved with a number of changes and

additions.

We agree with the reviewer, as this was largely due to the short form of the manuscript. We have significantly expanded the introduction in the revised manuscript.

Page 3, second paragraph:

It should be explained that in solution the dynamin tetramer exhibits basal GTP hydrolysis and that dynamin assembly into helices leads to stimulated GTP hydrolysis by promoting the G domain dimerization between tetramer subunits in adjacent helical rungs. The current description as written (“Dynamin exhibits assembly stimulated GTPase activity...”) is vague and doesn’t state where and how dimerization occurs in the context of the assembly or that it doesn’t occur in the context of the tetramer. The statement “Dimerization between GTPase domains is subject to the nucleotide states” is similarly ambiguous and should be clarified.

We have clarified the confusion and expanded the second paragraph by splitting it into two, first describes the dynamin monomers, second describing the interfaces. Please see the expanded introductions on Page 3-4.

Page 3, third paragraph:

A description of and comparison with the GTP-bound dynamin 1 reconstruction (Kong et al., 2018) should be included along with the GTP-bound K44A structure as it too forms a two-start helix that achieves a ‘super-constricted’ diameter and preceded the present study. It should also be noted somewhere that two-start assemblies have also been observed Dnm1 even with GMPPCP (Mears et al., 2011).

We appreciate the reviewer’s comment. We have compared representative cryoEM reconstructions of dynamin 1 in new Fig. S7. A discussion of all different reconstructions, including Dnm1, are added (Page 3-4). The diversity of helical symmetries of dynamin like proteins are further discussed on Page 10.

Page 4, first paragraph:

It would benefit the reader to explain how the super-constricted state relates to hemi-fission and why achieving such a narrow diameter is thought to be biologically and biophysically significant (see Bashkirov et al., 2008; Sundborger et al., 2014). Providing this connection will underscore why a two-start symmetry is significant.

We appreciate the reviewer’s comment and have further explained the relevance of two-start super-constricted state to dynamin function in the expanded introduction (Page 4), as well as in the Discussion on Page 10.

Adding a paragraph to the introduction that summarizes the current models for fission would help set the stage for the findings described in this work, especially since they are brought up again in the discussion.

We agree with the reviewer. The descriptions of existing models are included in introduction (Page 4). Based on our structure and previous structures from literatures, as well as our new results from elastic energy calculations, we proposed a new inclusive working model (Fig. 5f). We also added Fig. S7, which summarizes most representative cryo-EM reconstructions of dynamin 1 filaments, and included discussions in the light of these data (Page 10-11).

2) The second paragraph of the introduction enumerates all the domains, hinges, and interfaces that

form the building blocks of the dynamin polymer and play an important role in structural transitions. It would be helpful to include a supplementary figure that visually summarizes all these important features in place for unacquainted readers and less structurally inclined cell biologists.

We appreciate the reviewer's comment. Two new panels as Fig. 1a and Fig.1b has been added to provide a schematic of all the domains, hinges, interfaces. In Fig. 1a, constructs used in representative structural studies of full-length dynamin 1 (all with PRD domains truncated) are also shown.

3) Page 4: "We collected cryoEM of these dynamin tubes": It should be explicitly stated here that your reconstruction was carried out on MBP-tagged Δ PRD polymers in the absence of lipid. It is vague and confusing as written, as the previous sentence and figure legend references (Fig. 1f) refer to lipid bound tubes labeled with immuno-gold antibodies.

Modified according to the suggestion (now in Page 5).

4) Fig. 3: Inclusion of the one-start comparisons in panels 3C and 3D here seems out of place and would fit better with Fig. 4, especially since it referenced in the text when talking about the hinge 1 movements.

We agree with the reviewer. We re-organized our figures. Specifically, we moved the figure panels relating to the conservation of the GG interfaces to Fig. S3, included all residual-level analysis of the Stalk interfaces in Fig. S4. We combined the conformational analyses of both Hinge 1 and Hinge 2 of the one-start and the two-start dynamin filaments into one new figure (Fig. 3). Texts related to these changes are amended accordingly (Page 6-7).

5) Page 6, first paragraph: "In one start helix, the GG dimer interface is formed by the GTPase domains of the same strand". While this statement is technically true because it's only one single strand wrapping around, it's vague as written as it doesn't distinguish between dimers forming between adjacent helical rungs of the same strand versus adjacent neighboring subunits within the strand. It should be clarified to state that dimerization in a one-start helix only occurs between tetramers in the same strand following the completion of a helical turn. Additionally, either "helix" should be changed to "helices" or an "a" should be added to read "In a one start helix..."

Modified according to the suggestion in Page 6.

6) Discussion: It would be beneficial to expound upon the different models for dynamin catalyzed fission in the discussion and provide a more detailed description for exactly how the observations presented here add to the interpretation. A more direct comparison to the model suggested by Kong et al., 2018 describing how hydrolysis-dependent changes in the BSE and stalk could help drive constriction would be particularly warranted, especially since this incorporated a similar symmetry transition based on a lower-resolution two-start assembly.

As described above, we made new Fig. S7 to summarize major efforts in determining dynamin helical conformations. We also revised our model (Fig. 5f) to include the up-to-date mechanistic understandings of dynamin mediated membrane fission. The new theoretical elastic energy calculations in Fig. 5c-e provide rationale for two-start helical conformation and new mechanistic insights. The hydrolysis-dependent constriction and its relation to our structure solutions are discussed. Please see our significantly expanded discussion in Page 10-11.

Reviewer #2 (Remarks to the Author):

Dynamin-1 is a long-studied and fascinating GTPase involved in membrane fission reactions. Liu et al. report the $\sim 3.7\text{\AA}$ reconstruction a 2-start filament of polymeric dynamin-1. The structure is remarkably similar to the 1-start filament structure published in 2018, including the up/down asymmetry in hinge-2 centered near Ile289 and Arg290. Comparing the two morphologies shows us how very small rotations in the stalks, when summed in serial along the filament axis, lead to a dramatic change in the pitch of the helical assembly. This structural insight should be published--pending minor improvements.

We appreciate the reviewer for the nice summary of our work and positive feedback.

However, we recommend re-writing certain sections to be more cautious about the biological meaning of the 2-start helix. The functional model for fission via a 2-start structure remains unsubstantiated. The sample, moreover, has liabilities that limit insight into membrane fission mechanisms.

We have significantly expanded the Introduction (Page 3-4) and Discussion (Page 10-11). We have further added a new Fig. S7, which summarizes the major cryoEM structure determination efforts on dynamin filaments, and a new model Fig. 5f which is more inclusive for current understandings of dynamin function in the field. We also included discussions on sample limitations: 1) the MBP tag should have minimal impacts as described in Page 5 and Fig. 1e-h. We have made it clear that the sample contained an MBP tag. 2) the two-start dynamin filaments were assembled without lipid.

Recommendation:

To publish with softer functional claims, a broader discussion of the literature, and acknowledgment of the sample's limitations.

We appreciate the reviewer's comments. The functional relevance of the structural findings has been explored through the analysis of existing mutational data as shown in the new supplementary table 1., and the theoretical elastic energy calculations as illustrated in the new Fig. 5c-d (Page 8-9). A broader discussion of the literature and the current understanding of dynamin is included (Page 10-11) (also see the response to Reviewer #1's comment #6). For the sample limitations, please see the response above.

Major questions:

1. The paper is structure-only, without validating mutants tested in vitro or in cell experiments.

We appreciate the reviewer's comment. There are a wealth of biochemical and mutational data from the literature. We have made use of these existing data, to interpret the biological meaning of our structural findings, as summarized in the new Supplementary Table 1, in conjunction with the new Fig. S4. Furthermore, we have carried out analyses of elastic energies among assembly geometries, as shown in the new Fig. 5c-d. Please also see the response to the general comment of Reviewer #1.

Therefore, we advocate making the structural model as reliable as possible. The Rama-Z score provided by current versions of phenix reports a standard score for the overall Ramachandran space: the distribution of the (ϕ, ψ) torsion angles of the protein backbone. We mostly strive for "zero unexplained outliers", where explained outliers are justified by high-quality density. However, in the $\sim 3.5\text{\AA}$ -to- 4\AA resolution regimen, where the density is usually not sufficient to justify outliers, one can

overfit the model's ϕ and Ψ torsion angles. The Rama-Z quality metric measures the distance from the mean distribution to the observed distribution (how many standard deviations below or above the population mean a raw score is). This 2-start model falls in the Rama-Z "suspicious" region, suggesting that refinement software forced outlier residues into specific regions of the allowed, but not necessarily correct, dihedral space. We recommend the author's evaluate whether the model could be further improved before depositing the structures.

We appreciate the reviewer's expert and careful analysis of the map and the model. The high Rama-Z score for the initial coordinates comes from an earlier version of Phenix we used, which does not include the Rama-Z validations. We have now used a newer version (1.18.2-3874) of Phenix to carry out further refinements. The Rama-Z score now falls into the 'good' region. All statistics including the Rama-Z score are updated in the Table 1.

2. Related to #1, the clash score is high. We have seen this as well with filamentous samples, where clashes at the interface between subunits are common and challenging to address with current refinement algorithms. We hope further attention to model building will bring the clashscore down, but please still deposit the biological assembly as a helix (as you shared) rather than reduce the model to just a monomer or dimer (which would lead to better statistics, but a less useful pdb deposition for the community).

We thank the reviewer for this. We completely agree that depositing the biological assembly is a good practice to benefit the community, and the PDB deposited is indeed the helix. Further refinement as mentioned in response above to comment #1 has improved the clash score (from 28.68 to 13.01) of the helical assembly.

3. Hinge 2 and its asymmetry is a highlight of the paper. Please consider expanding the discussion of hinge 2 to include comparison of the 2-start with the 1-start reported by Kong et al. In the 2-start, this hinge appears to be a kink in the helix, rather than a loop between two helices? How would the authors interpret the Kong et al. observations regarding mutations in this region? Specifically, they report that a triple mutant that may boost the helical propensity near the Pro side of the kink (T292A/L293A/P294A) did result in a transferrin uptake deficiency--but the single mutant Pro294=>A, and a triple mutant closer to the residues highlighted here (R290A/D291A/T292A), was also fine for transferrin uptake?

We appreciate the reviewer's comment and have expanded the discussion of Hinge 2 to include comparison of the two-start with the one-start filaments (New Fig 3). The Hinge 2 in one monomer (green in Fig. 3) has the tendency to disfavour a helical conformation from residue I289 to P294, resulting in a kink. We added a discussion on the mutational data in Page 6-7.

Minor questions:

1. Include helical parameters used in table 1

Included.

2. Line 81 - please mention nucleotide state

Mentioned.

3. Line 139-140- If hinge 1 is so flexible, by what atomic mechanism would it communicate conformational change? Please hypothesize with data from the structure or remove this sentence.

We further analysed Hinge 1 and noticed its asymmetry in our two-start structure, even though the density is poorer for residues forming the Hinge. A new figure panel (Fig. 3d) is included and relevant descriptions are added on Page 6.

4. Fig S3. is busy and unlabeled. The message is difficult to appreciate.

We revised the figure (Fig. S4 now) to show clearly what residues are forming the interfaces and how well the interfaces are conserved between the one-start and the two-start helical configurations. We also introduced a new Supplementary Table 1 to summarize mutational data about these residues forming the Stalk interfaces. Furthermore, a new supplemental figure (Fig. S5) is drawn to show the conformational differences between dynamin in helical assemblies and dynamin in crystals. Relevant descriptions and discussions are added on Page 7-8.

5. Fig 5. It is difficult to keep track of the rotations.

Please consider illustrating where the membrane would be and the orientation of a larger helix with these views: a) membrane down, b, membrane down, but rotated, c) membrane behind page and d) membrane behind page, but rotated.

Incorporated the drawings of the membrane in Fig.4.

6. Please include a map-to-model FSC plot and filter the map to the appropriate resolution. Table suggests the model resolution is better than the half map resolution, which is unusual and hard to believe.

We apologize for the incorrect use of 0.143 cut-off for model resolution in the initial submission. According to the response to comment #1-2 above, further refinements are carried out. We included the map-to-model FSC as Fig S2c and updated the statistics in Table 1.

7. Consider expanding the discussion to acknowledge that this sample has liabilities that limit our ability to draw functional conclusions. These liabilities include i) the absence of a membrane and unconstrained PH domains, 2) a large MBP fusion on the N-terminus and 3) a deletion of the C-terminus.

For the limitations of the sample, 1) the absence of PH domain densities has been described on Page 6, 2) the reason and impacts of an MBP in the N-terminus has been described on Page 5, 3) the deletion of the C-terminus PRD domain has been necessary for all the high-resolution dynamin structures determined to date (Fig. 1a and Page 9).

References

- 1 Noel, J. K., Noé, F., Daumke, O. & Mikhailov, A. S. Polymer-like Model to Study the Dynamics of Dynamin Filaments on Deformable Membrane Tubes. *Biophysical Journal* **117**, 1870-1891, doi:<https://doi.org/10.1016/j.bpj.2019.09.042> (2019).
- 2 Ganichkin, O. *et al.* Quantification and demonstration of the constriction-by-ratchet mechanism in the dynamin molecular motor. *bioRxiv*, 2020.2009.2010.289546, doi:10.1101/2020.09.10.289546 (2021).
- 3 Sundborger, A. C. *et al.* A dynamin mutant defines a superconstricted prefission state. *Cell Rep* **8**, 734-742 (2014).

- 4 Kong, L. *et al.* Cryo-EM of the dynamin polymer assembled on lipid membrane. *Nature* **560**, 258-262, doi:10.1038/s41586-018-0378-6 (2018).

REVIEWER COMMENTS

Reviewer #1 (Remarks to the Author):

The revised manuscript by Liu et al. detailing the 3.74-Å cryo-EM structure of a two-start dynamin 1 filament is much improved, addressing many of the concerns raised by both reviewers. The authors have taken care to expand the introduction to provide better context and background for unacquainted readers and non-structural aficionados in the trafficking field and reorganized the figures in a more meaningful way. Additionally, the authors have added theoretical calculations and modeling of filament elastic energy, which complements the structural work and provides a possible mechanistic framework for understanding one-start and two-start dynamin helices. The findings presented herein will be an important benchmark for future structural studies. A few minor comments still should be addressed prior to publication.

1) Fig. 5e nicely illustrates the transition point in elastic energy for a one-start helix and could provide a direct mechanism for fission, wherein the energetic cost that accumulates as inner luminal diameter decreases could serve as a driving force for fission and disassembly (Figure 5f diagram, top). Does the elastic energy landscape change significantly for dimers within a two-start helices? This theoretical modeling is not presented and could be informative if it can be done.

Collectively, the calculations based on filament elastic energy argue that a two-start helix is actually a more energetically-favorable and more stable configuration of the dynamin polymer. It's somewhat unclear then, however, what the downstream mechanistic effects of a two-start helix would be. Would dynamin assemblies that achieve this configuration in vivo get trapped and have to disassemble before exerting further effects? Does this imply that one-start helices are in fact more biologically relevant for endocytosis and we don't capture them in the super-constricted state because the energetics would lead to rapid disassembly and/or scission events in vitro?

The cartoon in Figure 5f suggests that elastic stress could promote local defects and subsequent pitch expansion, leading to a two-start assembly, which in turn could lead to further constriction. How do the authors rectify this potential pathway with the fact that two-start assemblies have thus far only been observed in a super-constricted configuration that reaches the theoretical limit of membrane hemifission? Can a two-start configuration be achieved on a membrane template with wider inner luminal diameter (e.g. 7-10 nm) or is it constrained, either physically (by G domain interactions) or energetically?

It would be helpful to expand the discussion to discuss of these lingering questions.

2) Throughout the document and figure legends the authors use "protomer" and "monomer" interchangeably, often switching back and forth. It can be a bit confusing since protomer is never explicitly clarified and can mean different things in different structural contexts.

3) Figure 3: It would be helpful to include color-coded labeling of the individual structures superimposed in panels b-h and keep it consistent with the descriptions in the main text (see point 2, above). The figure legend is dense and detailed, making it difficult for the reader to sift through to have a clear sense of exactly what is being compared. The authors do a better job of labeling in Figure 2, where the individual domains within the tetramer are clearly marked.

4) Page 6-7, "The two Hinge conformations are asymmetric" section, second paragraph: It would be helpful to add "apo dynamin 1" to description of the 3SNH structure to read "...to a larger extent 34° when compared to the apo dynamin-1 crystal structure (PDB: 3SNH)." This is explicitly stated later and is important to specify explicitly what is being compared. Later, in the paragraph "PDB:" should be included in parentheses with the PDB code to read "...structure of dynamin 1 (PDB: 3SNH)7, Hinge 2 exhibits..."

5) There are a number of typos and grammatical errors that need to be fixed:

Page 3, second paragraph: should read "...Hinge 1 connects the BSE and the Stalk whereas Hinge 2 connects the BSE and the G domain"

Page 4, second paragraph: should read "...is critical for dynamin-mediated"

Page 5, last sentence of the introduction should be changed to present tense: "We discuss most existing structure solution (Fig. S7) and propose a working model consistent with the constriction/ratchet mechanism."

Page 5, first paragraph: The second sentence is awkwardly worded. Maybe change to: "This is probably due to lack of lipid constraints, which allows the dynamin filament to constrict into a more compact form."

Page 6, third paragraph, second sentence: should read "...we fitted the one-start GG dimer structure..."

Page 6, third paragraph, third sentence: should read either "24.43°, which corresponds to" or "24.43°, corresponding to..."

Page 8, first paragraph: "to note" should be changed to "noting" to read "it is worth noting that the mutagenesis analysis..."

Page 10, second paragraph, last sentence: change to present tense to read "Meanwhile, the asymmetry in Hinge 2 conformations likely confirms..."

Page 11, first paragraph, second sentence: should read "...described above (Fig. 5e), it emphasizes..."

Page 11, first paragraph, fifth sentence: should read "The GTP hydrolysis could result in..."

Page 11, first paragraph, 11th sentence: missing "An" to read "An in vitro reconstituted system..."

Figure 5, panel F: typos in "GTP powered constriction" and "elastic stress promotes local defect"

Reviewer #2 (Remarks to the Author):

The authors have addressed all of my concerns. However, I can't offer an informed opinion on the new elastic energy calculations. You may want to consider getting another reviewer with the right background to evaluate this new and interesting addition to the study.

Reviewer #3 (Remarks to the Author):

Liu et al. report the cryo-EM structure of a two-start dynamin filament at 3.74 Å resolution, providing new insight into the mechanism of membrane scission by dynamin. By solving the first high-resolution structure of the two-start dynamin filament, they demonstrated that small structural differences in the

GG dimer and the stalk tetramer of dynamin are sufficient for interconversion between the one-start and two-start helical filaments. Furthermore, they showed that the two-start helical filament is more stable than the one-start one from an elastic energy analysis. I support the publication of the manuscript after the following concerns are addressed properly.

1) In the discussion of working models of membrane fission by dynamin (Fig. 5f), it is not clear what is the effect of GTP hydrolysis. From the elastic energy analysis shown in Fig. 5c, it seems possible that the filament constricts with a larger pitch. And from the sentence on page 11 "The GTP hydrolysis could results in local disruptions of GG dimeric interfaces between different dynamin protomer within the one-start rung...", GTP hydrolysis seems to help the filament to take a larger pitch to allow the two-star helical filament. However, Fig. 5f seems inconsistent with the above view, showing that GTP energy is used to constrict the filament (and the membrane) with the same short pitch. Please elaborate on this point.

2) What are "the power-stroke like mechanisms of the GTPase-BSE motor" (page 10)?

Point by point responses to reviewers:

Reviewer #1 (Remarks to the Author):

The revised manuscript by Liu et al. detailing the 3.74-Å cryo-EM structure of a two-start dynamin 1 filament is much improved, addressing many of the concerns raised by both reviewers. The authors have taken care to expand the introduction to provide better context and background for unacquainted readers and non-structural aficionados in the trafficking field and reorganized the figures in a more meaningful way. Additionally, the authors have added theoretical calculations and modeling of filament elastic energy, which complements the structural work and provides a possible mechanistic framework for understanding one-start and two-start dynamin helices. The findings presented herein will be an important benchmark for future structural studies. A few minor comments still should be addressed prior to publication.

1) Fig. 5e nicely illustrates the transition point in elastic energy for a one-start helix and could provide a direct mechanism for fission, wherein the energetic cost that accumulates as inner luminal diameter decreases could serve as a driving force for fission and disassembly (Figure 5f diagram, top). Does the elastic energy landscape change significantly for dimers within a two-start helices? This theoretical modeling is not presented and could be informative if it can be done.

We thank the reviewer for pointing out this omission. We have added the plots of the two-start curve in Fig. 5e (red curve). In addition, it is important to note that the potential of shifting from one-start to two-start depends on the *relative* elastic energy between them. Thus, we have also added the elastic energy difference between one- and two-start helices to Fig. 5e (dashed green curve). In summary, we now show three curves in the new Fig. 5e and modified relevant discussions on Page 11.

Collectively, the calculations based on filament elastic energy argue that a two-start helix is actually a more energetically-favorable and more stable configuration of the dynamin polymer. It's somewhat unclear then, however, what the downstream mechanistic effects of a two-start helix would be. Would dynamin assemblies that achieve this configuration *in vivo* get trapped and have to disassemble before exerting further effects?

The elasticity energy analysis rationalizes why a dynamin two-start helix exists and how the conversion from one-start to two-start could happen. The reviewer asked a very interesting question about what happens after a two-start helix nucleates. We proposed two possibilities as shown in Fig. 5f (bottom right): 1) it may assemble into a longer two-start helix to remodel the membrane; 2) it may constrict actively to membrane scission. The first hypothesis (Constriction driven by further polymerization) is based on the fact that once a longer than one turn two-start helix formed, the geometry of dynamin filament would squeeze the membrane into a small diameter. The smallest inner diameter of a long two-start dynamin filament on top of a membrane tube observed *in vitro* is around 3.4 nm (Fig. S7e), which could spontaneously lead to membrane fission. However, we understand the complexity of *in vivo* conditions and also considered whether a long two-start helix with a wider diameter is able to actively constrict toward ultimate fission. We thus proposed the second hypothesis (GTP-dependent active constriction) that an active constriction within a two-start helix configuration is possible. To further support this, we carried out simulation work based on the previous model on how a one-start helix evolves in the presence of GTP¹. Starting with a two-start geometry, the result, similar to the one-start case, shows that the torques are non-cancelling, and thus, the rungs can rotate and generate constriction. We included the simulation as a new Supplementary Movie 1 and relative descriptions on Page 9-10 and discussions on Page 10-11. Therefore, it is unlikely that the dynamin two-start helix is a trap *in vivo*.

Does this imply that one-start helices are in fact more biologically relevant for endocytosis and we don't capture them in the super-constricted state because the energetics would lead to rapid disassembly and/or scission events *in vitro*?

As described above, the two-start helices could lead to membrane scission rather than a trap, and the super-constricted states observed in the two-start helices *in vitro* offered a simple hypothesis that a two-start helix could be the conformation right before the membrane scission. However, the elastic energy analysis also explained why one-start helices cannot or have not been captured in the super-constricted state *in vitro*. The minimum in the elastic energy of one-start helices is at radii greater than super-constricted. Assuming that one-start helices can reach tighter radii using the energy of GTP, this transient state can be very dynamic and likely leads to fast scission before being captured.

In sum, we do not intend to imply whether a specific configuration of dynamin helices is favoured biologically. Both one-start and two-start helices could lead to scission theoretically, and further investigations are required to differentiate their cellular impacts. We added a discussion on Page 12.

The cartoon in Figure 5f suggests that elastic stress could promote local defects and subsequent pitch expansion, leading to a two-start assembly, which in turn could lead to further constriction. How do the authors rectify this potential pathway with the fact that two-start assemblies have thus far only been observed in a super-constricted configuration that reaches the theoretical limit of membrane hemi-fission? Can a two-start configuration be achieved on a membrane template with wider inner luminal diameter (e.g. 7-10 nm) or is it constrained, either physically (by G domain interactions) or energetically?

We and others analyse well-ordered two-start helix *in vitro* for the purpose of structural determination, as shown in Fig. S7, the existence of other less well-ordered dynamin filaments, including two-start helices on a membrane template with wider inner luminal diameter, might be overlooked. More importantly, from our theoretical analysis, there are no geometric constraints that disallow a two-start helix with larger radius. The elastic energy could drive the two-start helix to a tighter one spontaneously *in vitro*. Meanwhile, a two-start helix with a larger diameter could constrict actively to the fission point with the power from GTP hydrolysis (Supplementary Movie 1). We added related discussions on Page 11-12.

It would be helpful to expand the discussion to discuss of these lingering questions.

Added according to the responses above.

2) Throughout the document and figure legends the authors use "protomer" and "monomer" interchangeably, often switching back and forth. It can be a bit confusing since protomer is never explicitly clarified and can mean different things in different structural contexts.

Changed all to "monomer".

3) Figure 3: It would be helpful to include color-coded labeling of the individual structures superimposed in panels b-h and keep it consistent with the descriptions in the main text (see point 2, above). The figure legend is dense and detailed, making it difficult for the reader to sift through to have a clear sense of exactly what is being compared. The authors do a better job of labeling in Figure 2, where the individual domains within the tetramer are clearly marked.

We have edited the figure, introduced colour code for labelling dynamin monomers in an additional panel Figure 3i, and simplified the legend.

4) Page 6-7, "The two Hinge conformations are asymmetric" section, second paragraph:

It would be helpful to add “apo dynamin 1” to description of the 3SNH structure to read “...to a larger extent 34° when compared to the apo dynamin-1 crystal structure (PDB: 3SNH).” This is explicitly stated later and is important to specify explicitly what is being compared. Later, in the paragraph “PDB:” should be included in parentheses with the PDB code to read “...structure of dynamin 1 (PDB: 3SNH)7, Hinge 2 exhibits...”

Modified.

5) There are a number of typos and grammatical errors that need to be fixed:

Appreciate for pointing out.

Page 3, second paragraph: should read “...Hinge 1 connects the BSE and the Stalk whereas Hinge 2 connects the BSE and the G domain”

Modified.

Page 4, second paragraph: should read “...is critical for dynamin-mediated”

Modified.

Page 5, last sentence of the introduction should be changed to present tense: “We discuss most existing structure solution (Fig. S7) and propose a working model consistent with the constriction/ratchet mechanism.”

Modified.

Page 5, first paragraph: The second sentence is awkwardly worded. Maybe change to: “This is probably due to lack of lipid constraints, which allows the dynamin filament to constrict into a more compact form.”

Modified.

Page 6, third paragraph, second sentence: should read “...we fitted the one-start GG dimer structure...”

Modified.

Page 6, third paragraph, third sentence: should read either “24.43°, which corresponds to” or “24.43°, corresponding to...”

Modified.

Page 8, first paragraph: “to note” should be changed to “noting” to read “it is worth noting that the mutagenesis analysis...”

Modified.

Page 10, second paragraph, last sentence: change to present tense to read “Meanwhile, the asymmetry in Hinge 2 conformations likely confirms...”

Modified.

Page 11, first paragraph, second sentence: should read “...described above (Fig. 5e), it emphasizes...”

Modified.

Page 11, first paragraph, fifth sentence: should read “The GTP hydrolysis could result in...”

Modified.

Page 11, first paragraph, 11th sentence: missing “An” to read “An in vitro reconstituted system...”

Modified.

Figure 5, panel F: typos in “GTP powered constriction” and “elastic stress promotes local defect”
Modified.

Reviewer #2 (Remarks to the Author):

The authors have addressed all of my concerns. However, I can't offer an informed opinion on the new elastic energy calculations. You may want to consider getting another reviewer with the right background to evaluate this new and interesting addition to the study.

We appreciate the reviewer's comment.

Reviewer #3 (Remarks to the Author):

Liu et al. report the cryo-EM structure of a two-start dynamin filament at 3.74 Å resolution, providing new insight into the mechanism of membrane scission by dynamin. By solving the first high-resolution structure of the two-start dynamin filament, they demonstrated that small structural differences in the GG dimer and the stalk tetramer of dynamin are sufficient for interconversion between the one-start and two-start helical filaments. Furthermore, they showed that the two-start helical filament is more stable than the one-start one from an elastic energy analysis. I support the publication of the manuscript after the following concerns are addressed properly.

1) In the discussion of working models of membrane fission by dynamin (Fig. 5f), it is not clear what is the effect of GTP hydrolysis. From the elastic energy analysis shown in Fig. 5c, it seems possible that the filament constricts with a larger pitch. And from the sentence on page 11 “The GTP hydrolysis could results in local disruptions of GG dimeric interfaces between different dynamin protomer within the one-start rung...”, GTP hydrolysis seems to help the filament to take a larger pitch to allow the two-star helical filament. However, Fig. 5f seems inconsistent with the above view, showing that GTP energy is used to constrict the filament (and the membrane) with the same short pitch. Please elaborate on this point.

The role of GTP hydrolysis is to drive the GTPase cycle of the G-domain/BSE. This involves the repeated formation and dissociation of neighbouring GTPase domains. Hydrolysis itself generates a force between neighbouring turns. Binding of GTP or GDP. AlF_4^- appears to favour a strong GG dimer. An important consequence of the presence of GTP and the dimerization it promotes is that neighbouring turns tend to closely interact. This could be the reason that a one-start helix may be possible despite being higher in elastic stress. The upper route of the Fig. 5f depicts this view of dynamin's operation.

The bottom route and the SI movie explores the possible role of two-start helices during dynamin constriction. While it is indeed the case that GTP promotes dimerization, the cyclical nature of the GTPase cycle means that there are states that are not favourable for dimerization (GDP and apo). Given the increased tendency for pitch expansion under tight constriction, breaks in the connection between neighbouring turns may be sufficiently long-lived to nucleate a two-start helix (Fig.5f bottom route).

To address the issues mentioned by the reviewer, we have significantly revised the sentences on mentioned by the reviewer on Page 11. To better describe the effect of GTP and the GTPase cycle,

we also added one sentence on Page 3 to introduce the GTPase cycle, and additional explanations in the discussion on Page 10-11 where the one-start constriction mechanism is introduced.

2) What are “the power-stroke like mechanisms of the GTPase-BSE motor” (page 10)?

Modified relative sentences and added a reference on Page 10.

Reference

- 1 Ganichkin, O. M. *et al.* Quantification and demonstration of the collective constriction-by-ratchet mechanism in the dynamin molecular motor. *Proc Natl Acad Sci U S A* **118**, doi:10.1073/pnas.2101144118 (2021).

REVIEWERS' COMMENTS

Reviewer #1 (Remarks to the Author):

The authors have sufficiently addressed my comments and I support publication of this work.

Reviewer #3 (Remarks to the Author):

The authors have addressed all of my concerns properly in the revised manuscript.